# COPI mediates recycling of an exocytic SNARE by recognition of a ubiquitin sorting signal

Peng Xu[1], Hannah M Hankins[1], Chris MacDonald[2], Samuel J Erlinger[1], Meredith N Frazier[1], Nicholas S Diab[1], Robert C Piper[2], Lauren P Jackson[1], Jason A MacGurn[3], Todd R Graham[1]*

[1]Department of Biological Sciences, Vanderbilt University, Nashville, United States; [2]Department of Molecular Physiology and Biophysics, University of Iowa, Iowa City, United States; [3]Department of Cell and Developmental Biology, Vanderbilt University Medical Center, Nashville, United States

**Abstract** The COPI coat forms transport vesicles from the Golgi complex and plays a poorly defined role in endocytic trafficking. Here we show that COPI binds K63-linked polyubiquitin and this interaction is crucial for trafficking of a ubiquitinated yeast SNARE (Snc1). Snc1 is a v-SNARE that drives fusion of exocytic vesicles with the plasma membrane, and then recycles through the endocytic pathway to the Golgi for reuse in exocytosis. Removal of ubiquitin from Snc1, or deletion of a β'-COP subunit propeller domain that binds K63-linked polyubiquitin, disrupts Snc1 recycling causing aberrant accumulation in internal compartments. Moreover, replacement of the β'-COP propeller domain with unrelated ubiquitin-binding domains restores Snc1 recycling. These results indicate that ubiquitination, a modification well known to target membrane proteins to the lysosome or vacuole for degradation, can also function as recycling signal to sort a SNARE into COPI vesicles in a non-degradative pathway.
DOI: https://doi.org/10.7554/eLife.28342.001

*For correspondence:
tr.graham@vanderbilt.edu

Competing interests: The authors declare that no competing interests exist.

## Introduction

Sequential rounds of vesicle budding and fusion reactions drive protein transport through the secretory and endocytic pathways (*Rothman, 1994*). Vesicle budding often requires cytosolic coat proteins, such as COPI, COPII or clathrin, that assemble onto the donor organelle to mold the membrane into a tightly curved structure while collecting cargo for inclusion into the nascent vesicle (*Faini et al., 2013*). Efficient departure of cargo from the donor organelle requires a sorting signal within the cargo protein that is recognized by the coat complexes. For example, the heptameric COPI coat complex assembles onto Golgi membranes where it selects cargo bearing a C-terminal di-lysine (KKxx-COO⁻ or KxKxx-COO⁻) sorting signal exposed on the cytosolic side of the membrane (*Eugster et al., 2004*; *Letourneur et al., 1994*; *Waters et al., 1991*). The two large COPI subunits, α- and β'-COP, each contain a pair of WD40 repeat domains that form twin β-propellers used to select cargo by binding to the sorting signal (*Jackson, 2014*; *Jackson et al., 2012*). After budding, COPI vesicles uncoat and deliver the di-lysine bearing cargo to the ER by fusing to this acceptor membrane in a SNARE-dependent reaction (*Rein et al., 2002*; *Südhof and Rothman, 2009*).

Intrinsic to the SNARE hypothesis is the privileged selection of v-SNAREs by the vesicle budding machinery to ensure the nascent vesicle can fuse to its target membrane bearing complementary t-SNAREs, and the subsequent need for v-SNARE recycling back to the donor compartment (*Miller et al., 2011*). v-SNAREs are small, single-pass membrane proteins with the C-terminus embedded within the lumenal space (*Weber et al., 1998*). Thus, COPI cannot select v-SNAREs using

the canonical C-terminal motifs and there is no well-defined sorting signal within a v-SNARE that is recognized by COPI. A further complication with the vesicular transport process is that it leads to depletion of v-SNAREs from the donor membrane and their deposition in the acceptor membrane. Thus, it is essential to recycle the v-SNAREs back to the donor compartment in order to sustain the vesicular transport pathway. Exocytic v-SNAREs that bud from the trans-Golgi network and target vesicles to the plasma membrane have served as models to understand the mechanisms of SNARE recycling, although these studies have primarily focused on how the v-SNAREs are endocytosed from the plasma membrane for delivery to early endosomes (*Burston et al., 2009*; *Lewis et al., 2000*; *Miller et al., 2011*). The subsequent step of transport from endosomes back to the Golgi, however, is poorly understood.

The yeast endocytic pathway is complex and tightly coupled to the Golgi through several vesicular transport pathways. Proteins and lipids internalized from the plasma membrane can be targeted to separate compartments marked by the syntaxin (t-SNARE) homologs Pep12 and Tlg1 (*Holthuis et al., 1998a*, *1998b*; *Prescianotto-Baschong and Riezman, 2002*; *Wiederkehr et al., 2000*). Tlg1 marks an organelle where bulk-phase endocytic tracers (cationized nanogold) first appear after internalization from the plasma membrane, and therefore this organelle has been described as an early endosome (*Prescianotto-Baschong and Riezman, 2002*). Endocytosed proteins that rapidly recycle back to the plasma membrane (for example, the exocytic v-SNARE Snc1 and the chitin synthase Chs3) also enter the Tlg1 compartment via endocytosis, whereas internalized cargo targeted for destruction in the vacuole may be transported directly to Pep12-marked endosomes (also called the prevacuolar compartment) (*Holthuis et al., 1998b*). Therefore, Tlg1 may mark an early/recycling endosome and is required for Snc1 recycling as GFP-Snc1 accumulates in internal compartments in *tlg1Δ* cells (*Lewis et al., 2000*). However, the distinction between the TGN and Tlg1-positive early endosome is blurred as a portion of Tlg1 also localizes to a Sec7-marked compartment considered the yeast TGN, and TGN resident proteins can also be found in the Tlg1 compartment (*Holthuis et al., 1998a*; *Holthuis et al., 1998b*; *McDonald and Fromme, 2014*; *Prescianotto-Baschong and Riezman, 2002*). In spite of the similarity in protein composition, the early/recycling endosome appears to be functionally distinct from the TGN as mutants defective in Snc1 recycling (e.g. *rcy1Δ*) trap GFP-Snc1 in an enlarged Tlg1-positive compartment that is deficient for the TGN marker Sec7 (*Furuta et al., 2007*; *Lewis et al., 2000*). Newly synthesized proteins are secreted from *tlg1Δ* or *rcy1Δ* mutants with wild-type kinetics implying formation of exocytic vesicles at the TGN is unperturbed (*Holthuis et al., 1998a*; *Wiederkehr et al., 2000*), and therefore the GFP-Snc1 recycling defect is thought to occur at a vesicular transport step between an early/recycling endosome and TGN. However, given the uncertainty in the nature of these compartments marked by Tlg1 and Sec7 (early endosome, TGN, or a hybrid of these organelles), we use the term 'recycling' to indicate movement of GFP-Snc1 from the endocytic pathway to the exocytic pathway. Here, we sought to clarify the sorting signals in Snc1-GFP and vesicle coat protein acting in this recycling step.

Snc1 recycling is independent of retromer and clathrin adaptors known to mediate transport of other cargos in these pathways (*Lewis et al., 2000*). Instead, an F-box protein (Rcy1) (*Galan et al., 2001*), a phosphatidylserine flippase (Drs2/Cdc50) (*Furuta et al., 2007*; *Hua et al., 2002*; *Xu et al., 2013*), an ArfGAP (Gcs1) (*Robinson et al., 2006*), and a sorting nexin complex (Snx4/41) (*Hettema et al., 2003*; *Ma et al., 2017*) are required for recycling of Snc1, although the precise functions for these proteins remain unclear. F-box proteins are best known as substrate-selecting adaptors in Skp1-Cullin-F-box (SCF) E3 ubiquitin (Ub) ligases, but the Rcy1-Skp1 complex plays a role in Snc1 recycling that is independent of the cullin subunit or the Cdc34 E2 Ub ligase (*Galan et al., 2001*). Moreover, ubiquitination of membrane proteins in the endocytic pathway is thought to set a course for their degradation in the lysosome or vacuole via the ESCRT/MVB pathway (*MacGurn et al., 2012*). Thus, it seemed unlikely that Rcy1 mediates ubiquitination of Snc1 in order to recycle this SNARE protein out of the endocytic pathway. Nonetheless, several high-throughput studies have shown that Snc1 is ubiquitinated (*Peng et al., 2003*; *Silva et al., 2015*; *Swaney et al., 2013*), and altering a targeted lysine to arginine (Snc1-K63R) surprisingly perturbed its recycling (*Chen et al., 2011*), suggesting ubiquitin (Ub) conjugation could play a role in this trafficking pathway.

Here we provide evidence that polyubiquitin (polyUb) chains are indeed a sorting signal that drives recycling of Snc1, and surprisingly find that COPI mediates this sorting event by direct binding

to K63-linked polyUb chains. COPI was observed to localize to early endosomes in mammalian cells more than two decades ago (*Aniento et al., 1996*; *Whitney et al., 1995*), and was initially thought to mediate transport of proteins from early endosomes to late endosomes in animal cells (*Aniento et al., 1996*). However, as models for early to late endosome maturation emerged (*Scott et al., 2014*; *Zerial and McBride, 2001*), this proposed role for endosomal COPI was abandoned and there remains no clearly defined role for COPI in protein trafficking through the endosomal system. A major impediment to deciphering a function for endosomal COPI is that mutations or knockdown approaches that inactivate COPI grossly disrupt Golgi function. Thus, any endosomal defects observed in COPI-deficient cells could be attributed to an indirect downstream effect of perturbing the Golgi complex. We describe here a set of COPI separation-of-function mutations and fusion proteins that indicate a direct role of this coat in selecting Snc1 bearing K63-linked Ub chains for recycling.

## Results

### Snc1 ubiquitination is required for recycling

In wild-type (WT) cells, nearly half of GFP-Snc1 localizes to the plasma membrane (preferentially at the buds relative to the mother cell) and the remainder is observed in cytoplasmic punctae (*Figure 1A,B*). As previously shown (*Lewis et al., 2000*), deletion of RCY1 (rcy1Δ) caused GFP-Snc1 accumulation in punctae and a loss from the plasma membrane (*Figure 1A,B*). Snc1 is ubiquitinated and a lysine mutation that reduces ubiquitination also caused GFP-Snc1 accumulation in punctae marked by the endocytic tracer FM4-64 (*Chen et al., 2011*), suggesting that Ub could be a retrieval signal. However, lysines are commonly found in protein sorting signals, undergo a variety of other post-translational modifications, and serve structural roles, so it was unclear whether it was the loss of lysine or ubiquitination per se that caused the Snc1 recycling defect.

To address whether ubiquitination is required for Snc1 recycling, we fused the UL36 deubiquitinase (DUB) from Herpes simplex virus (*Stringer and Piper, 2011*), as well as a catalytically inert mutant (DUB*), to GFP-Snc1. This DUB can effectively strip Ub from a fusion partner without altering the amino acid sequences targeted for ubiquitination (*Stringer and Piper, 2011*). In contrast to GFP-Snc1, DUB-GFP-Snc1 localized to punctae marked by Tlg1. In addition, DUB-GFP-Snc1 mislocalized to punctae in WT cells to the same extent as GFP-Snc1 mislocalized to punctae in rcy1Δ cells (*Figure 1A–D*). Deubiquitinase activity was required to block recycling as DUB*-GFP-Snc1 localized normally to the plasma membrane (*Figure 1A–D*). To determine if localization of DUB-GFP-Snc1 to internal punctae required endocytosis, we mutated the Snc1 endocytic internalization signal in the context of the DUB and DUB* fusion proteins (*Lewis et al., 2000*). The endocytosis-defective variants (e.g. DUB-GFP-Snc1-PM) accumulated at the plasma membrane (*Figure 1A–D*), suggesting that the DUB delayed movement from endosomes to the TGN rather than TGN to plasma membrane transport when attached to WT Snc1.

It was possible that the DUB interfered with Snc1 trafficking by deubiquitinating the trafficking machinery required for recycling rather than solely deubiquitinating Snc1 itself. As a further test for the specificity of the DUB block in Snc1 recycling, we also attached DUB and DUB* to Drs2, an integral membrane phosphatidylserine flippase that localizes to the TGN and early endosomes, and is part of the machinery required for recycling of GFP-Snc1(*Chen et al., 1999*; *Hua et al., 2002*). Whereas GFP-Snc1 accumulated in punctae within drs2Δ cells, recycling to the plasma membrane was fully restored in drs2Δ cells expressing Drs2-DUB or Drs2-DUB* (*Figure 1E–F*). Thus, attaching DUB to a component of the trafficking machinery in this pathway had no effect on Snc1 recycling, suggesting that the DUB is not significantly acting on neighboring proteins within this pathway.

### Snc1 is extensively modified with polyUb chains

A prior study identified a mono-ubiquitinated form of Snc1 (*Chen et al., 2011*), which we confirmed, but we also found evidence for Snc1 forms heavily modified with polyUb (*Figure 2A–B*). The observation that Snc1 ubiquitination is significantly reduced in yeast strains expressing K63R ubiquitin as the sole source of Ub, and thus cannot generate K63-linked polyUb chains (*Silva et al., 2015*) supports this conclusion. Rcy1 is required for Snc1 recycling and was implicated in Snc1 ubiquitination. Therefore, we tested whether DUB fusion to this F-box protein would perturb GFP-Snc1 recycling.

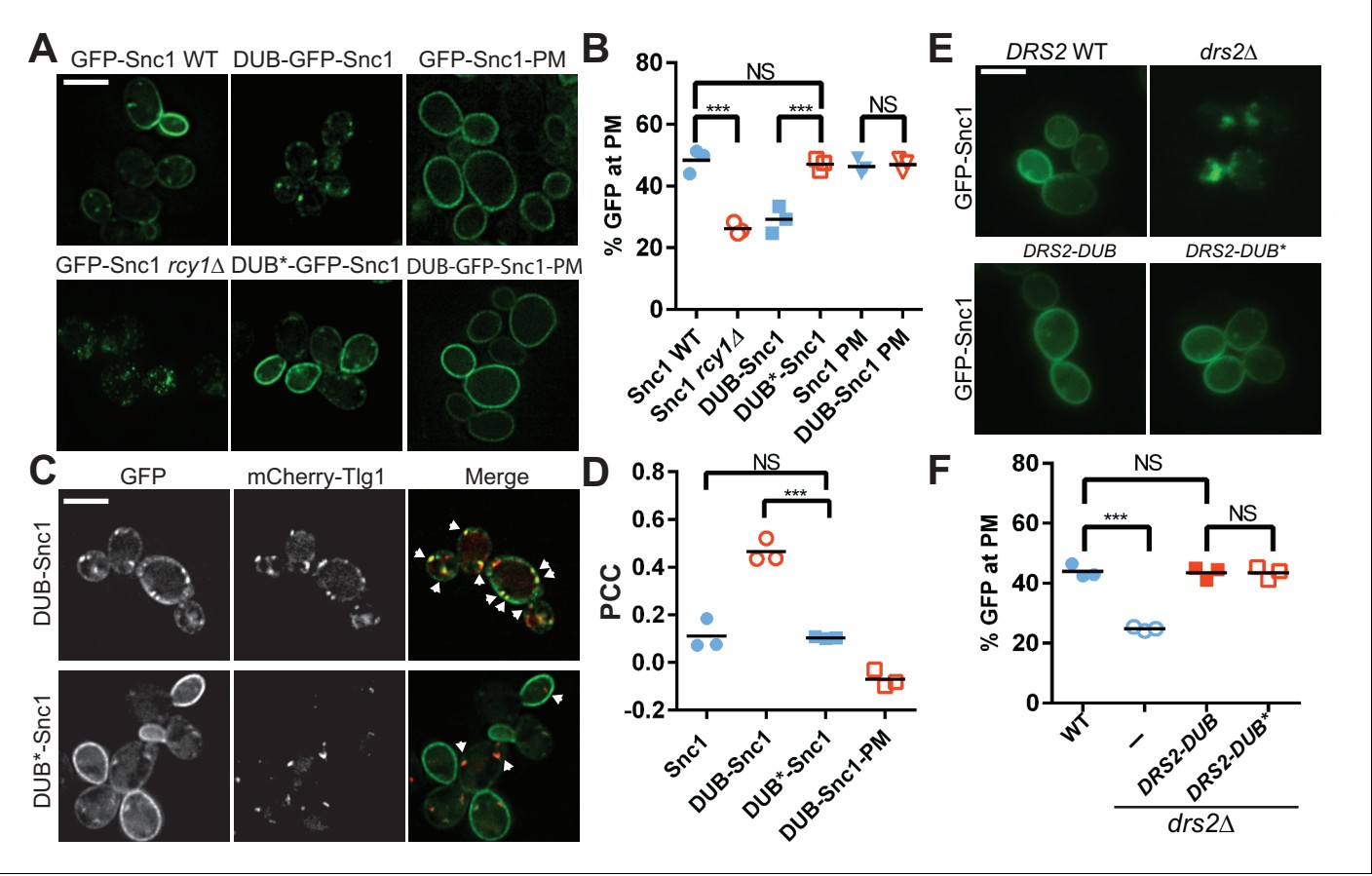

**Figure 1.** Ubiquitination is required for Snc1 recycling. (A) Fusion of catalytically active deubiquitinase (DUB), but not the inactive form (DUB*) to GFP-Snc1 blocks its recycling comparably to *rcy1Δ*. Mutation of an endocytic signal (PM) in Snc1 prevents accumulation of DUB-GFP-Snc1-PM in cytosolic punctae. (B) Quantification of GFP intensity at the plasma membrane. At least 50 cells for three biological replicates of each genotype were analyzed, and the value and mean for each biological replicate were plotted. (C) DUB-GFP-Snc1 accumulates in punctae marked by mCherry-Tlg1. The arrowheads highlighted the punctae showing colocalized GFP-Snc1 with mCherry-Tlg1. (D) Pearson correlation coefficient (PCC) GFP-Snc1 with mCherry-Tlg1. Each biological replicate plotted includes at least 20 cells. (E) Fusion of DUB to Drs2 does not disrupt the ability of Drs2 to support Snc1 recycling. (F) Quantification of GFP intensity at the plasma membrane for the cells in (E). Each biological replicate includes at least 50 cells. Statistical differences in (B), (D) and (E) were determined using a one-way ANOVA on the means of three biological replicates (***p<0.001; NS, p>0.05). Scale bar in (A), (C) and (E) represents 5 μm.

DOI: https://doi.org/10.7554/eLife.28342.002

The following source data is available for figure 1:

**Source data 1.** This spreadsheet contains the three means of GFP intensity at the plasma membrane data used to generate the dot plots shown in *Figure 1B and F*, and the three means of Pearson correlation coefficient data used to generate the dot plots shown in *Figure 1D*.

DOI: https://doi.org/10.7554/eLife.28342.003

Attachment of DUB to the N-terminus of Rcy1 (as the sole source of this protein) did cause a partial defect in GFP-Snc1 recycling; however, the DUB* fusion protein caused the same partial defect. Therefore, the DUB-Rcy1 fusions partially disrupted Rcy1 function by a mechanism that is independent of deubiquitinase activity (*Figure 2C,D*), which further emphasizes the specificity of the effect of DUB when fused to GFP-Snc1.

To identify other ligases potentially acting on Snc1, we screened through a collection of E3 ligase-DUB fusion proteins (*MacDonald et al., 2017*) to see if any disrupted GFP-Snc1 recycling. DUB fusions with two endosome-localized E3 Ub ligases, Pib1 and Tul1, strongly perturbed GFP-Snc1 recycling (*Figure 2E*), while DUB fusion with the Rsp5 and Vps11 E3 ligases were without effect. Moreover, we immunoprecipitated untagged Snc1 from cells expressing Pib1-DUB and found that ubiquitination of Snc1 was reduced relative to Snc1 from the control strain (*Figure 2F*). We then

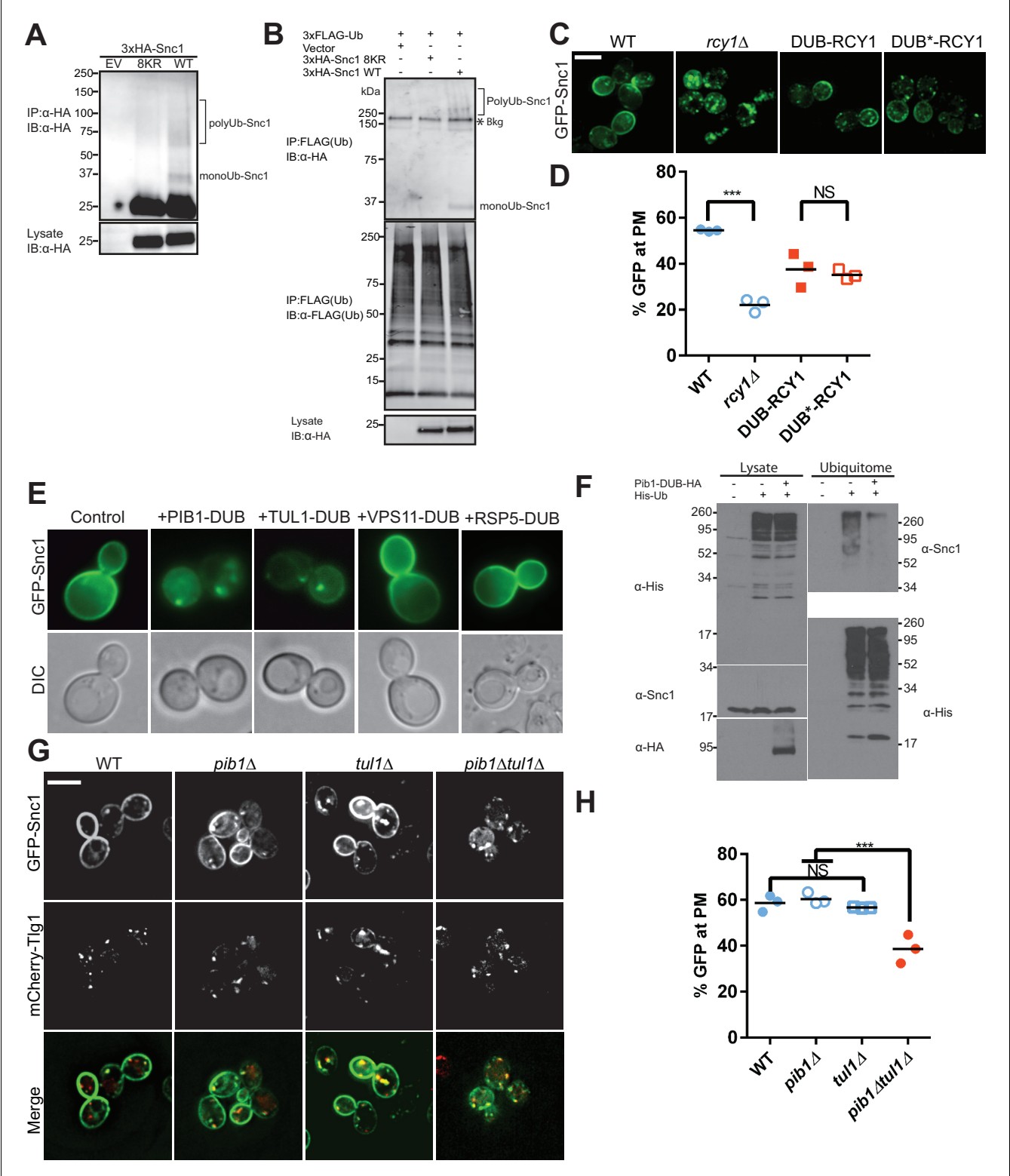

**Figure 2.** Snc1 is extensively modified with polyUb chains. (A) Proteins were immunoprecipitated from strains expression an empty vector, a lysineless 3XHA-Snc1-8KR, or WT 3XHA-Snc1 with anti-HA beads, and then immunoblotted using the anti-HA antibody. Monoubiquitinated and polyubiquitinated Snc1 forms are indicated (based on predicted mobilities). (B) Ubiquitinated proteins were immunoprecipitated using anti-FLAG antibodies from strains expressing 3X-FLAG-Ub and either an empty vector, a lysineless 3XHA-Snc1-8KR, or WT 3XHA-Snc1, and then immunoblotted for HA-Snc1 or FLAG-Ub. Monoubiquitinated and polyubiquitinated Snc1 forms are indicated (based on predicted mobilities). The lysine-less 3XHA-
*Figure 2 continued on next page*

Figure 2 continued

Snc1-8KR is a specificity control and the asterisk indicates background bands. (C) Rcy1 appears to play a role in Snc1 recycling that is independent of Ub ligase activity. A fusion of DUB or DUB* to the amino terminus of Rcy1 caused a partial defect in Snc1 recycling when expressed in rcy1Δ cells (BY4742 YJL204C). However, there was no significant difference between DUB and DUB*, indicating that the effect of the DUB is unrelated to its deubiquitinase activity. (D) Quantification of GFP intensity at the plasma membrane. (E) WT cells (BY4742) overexpressing GFP-Snc1 and DUB fusions with several candidate E3 Ub ligases. PIB1-DUB and TUL1-DUB were the only ligase-DUB fusions that caused a GFP-Snc1 recycling defect. (F) DUB tagged Pib1 significantly reduced endogenous polyubiquitinated Snc1. Ubiquitinated proteins (Ubiquitome) were recovered from cells expressing His-tagged Ub with or without Pib1-DUB and probed for endogenous, untagged Snc1 and His-Ub. Much less polyubiquitinated Snc1 was recovered from the ubiquitome in cells expressing DUB-Pib1. DUB-Pib1-HA expression was confirmed by immunoblot with anti-HA antibody. (G) The pib1Δ (PLY5293) and tul1Δ (PLY5294) single mutants recycled GFP-Snc1 normally, but the pib1Δ tul1Δ (PXY64) double mutant displayed a recycling defect. (H) Quantification of GFP intensity at the plasma membrane for cells shown in (G). Each biological replicate includes at least 50 cells for data plotted in (D) and (H). Statistical differences were determined using a one-way ANOVA on the means of three biological replicates. (***$p < 0.001$; NS, $p > 0.05$). Scale bar in (C) and (G) represents 5 μm.

DOI: https://doi.org/10.7554/eLife.28342.004

The following source data and figure supplements are available for figure 2:

**Source data 1.** This spreadsheet contains the three means of GFP intensity at the plasma membrane of Rcy1 mutant cells and pib1, tul1 mutant cells used to generate the dot plots shown in *Figure 1D and H*.
DOI: https://doi.org/10.7554/eLife.28342.007
**Figure supplement 1.** Inactivation of a COPI temperature-sensitive allele (ret1-1) at the non-permissive temperature (37°C) blocked GFP-Snc1 recycling.
DOI: https://doi.org/10.7554/eLife.28342.005
**Figure supplement 2.** Plasma membrane proteins Ina1 and Tat1 were tagged with mNeonGreen and visualized in WT (wild-type) and pib1Δ tul1Δ double mutant cells by fluorescence microscopy.
DOI: https://doi.org/10.7554/eLife.28342.006

tested the pib1Δ and tul1Δ single mutants, which were without phenotype, but the pib1Δ tul1Δ double mutant displayed a GFP-Snc1 recycling defect (*Figure 2G,H*). The localization of two other plasma membrane proteins examined, Tat1 and Ina1, were unaffected in the pib1Δ tul1Δ mutant (*Figure 2—figure supplement 2*). These results suggest that Pib1 and Tul1, rather than Rcy1, are primarily responsible for Snc1 ubiquitination.

## COPI binds K63-linked polyUb directly and this interaction is required for GFP-Snc1 recycling

Given that Snc1 ubiquitination is critical for its recycling, we hypothesized that Ub conjugated to Snc1 may function as a sorting signal in the recycling pathway and we asked what trafficking machinery might recognize Ub as a sorting determinant. Inactivation of COPI using a temperature conditional allele (ret1-1) blocked GFP-Snc1 recycling, whereas clathrin adaptor mutants and retromer mutants had no effect (*Figure 2—figure supplement 1*) (*Lewis et al., 2000*; *Robinson et al., 2006*). However, the Golgi is markedly perturbed when COPI is inactivated, undermining clear interpretation of this result due to possible indirect effects of disrupting the Golgi. If COPI plays a direct role in Ub-dependent Snc1 recycling, we reasoned it might bind the Ub sorting signal and the endosomal Snc1 recycling function should be independent of established COPI functions at the Golgi complex.

COPI is a heptamer composed of a B-subcomplex (α/β'/ε-COP subunits) structurally similar to clathrin heavy chains (*Figure 3A*), and an F-subcomplex similar to tetrameric clathrin adaptors (not shown) (*Dodonova et al., 2015*; *Faini et al., 2013*; *Fiedler et al., 1996*). The α and β'-COP subunits in the B-subcomplex each have two WD40 repeat propeller domains at their N-termini that bind dilysine sorting motifs. All well-characterized sorting signals recognized by COPI are near the C-terminus of the cargo and the N-terminal propellers of α and β'-COP use a basic patch to coordinate the carboxyl group (*Jackson, 2014*). However, many WD40 repeat domains also bind Ub (*Pashkova et al., 2010*). Therefore, we examined COPI WD40 propeller domains for interaction with Ub and found that they bound to K63-linked tetraUb (K63 Ub$_4$) (*Figure 3B*). The N-terminal propeller of β'-COP (1-304) bound slightly better to Ub$_4$ than the first propeller of α-COP (1-327), and a fragment of β' carrying both propellers (1-604) bound most efficiently.

Binding of β'-COP propellers to Ub was remarkably specific for linkage and chain-length. Polymers containing 5 or more K63-linked Ubs were required for the most robust and specific binding (*Figure 3C*). In contrast, β'-COP (1-304) did not bind significantly to K48-linked Ub chains or mono-Ub (*Figure 3C* and *Figure 3—figure supplement 1*). For the latter experiment, we incubated [15]N-

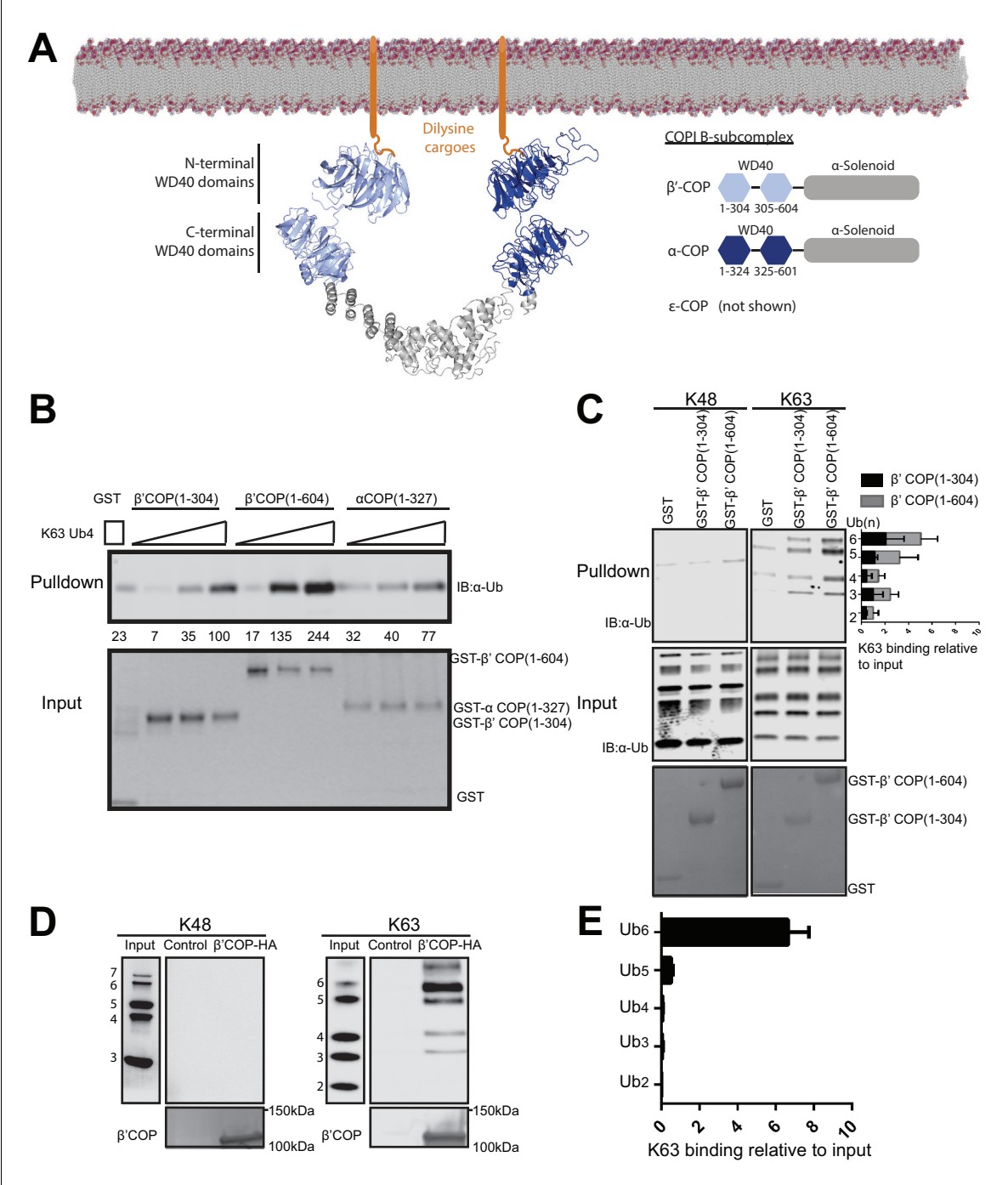

**Figure 3.** WD40 repeat propeller domains of COPI bind K63-linked polyubiquitin. (**A**) Structures of α- and β'-COP from the COPI B-subcomplex shown binding dilysine cargo in the membrane. (**B**) GST-β'COP (1-604), GST-β'COP (1-304) and GST-αCOP (1-327) bind K63-linked tetraUb relative to the GST only control. 0.5 µM of GST and GST tagged WD40 proteins immobilized glutathione beads were incubated 125 nM, 250 nM or 500 nM of K63-Ub₄. Values are tetraUb band intensities from this experiment. (**C**) Both GST-β'COP (1-604) and GST-β'COP (1-304) preferentially binds long K63-linked chains of Ub. Quantification of K63 binding relative to input (100*(band signal intensity – corresponding GST lane)/input band intensity). The values represent mean ± SEM from three independent binding experiments. (**D**) COPI isolated from yeast on anti-HA beads also preferentially binds long K63-linked polyUb, but not K48-linked Ub. (**E**) Quantification of K63-linked Ub polymers binding relative to input. The values represent mean ± SEM from three independent binding experiments (100*band signal intensity/input band intensity).

DOI: https://doi.org/10.7554/eLife.28342.008

The following figure supplement is available for figure 3:

**Figure supplement 1.** The N-terminal propeller of β'-COP (1-304) does not bind significantly to monoUb.

*Figure 3 continued on next page*

*Figure 3 continued*

DOI: https://doi.org/10.7554/eLife.28342.009

ubiquitin with or without a 10-fold molar excess of β'-COP (1-304) and observed no difference in the HSQC NMR spectra, indicating no measurable interaction under these conditions (*Figure 3—figure supplement 1*). Moreover, short K63-linked polymers bound poorly to β'-COP (1-304) in the GST-pulldown, competitive binding experiments (Ub$_2$-Ub$_4$) (*Figure 3C,D*). β'-COP (1-604), with both pro-pellers, retained the same specificity but recovered greater amounts of polyUb in these assays (*Figure 3C,D*). These results demonstrate that the β'-COP and α-COP propeller domains can directly and specifically bind K63-linked poly-Ub.

To be certain that these Ub interactions were not an artifact of the recombinant GST fusion pro-tein fragments assayed, we tested if the COPI complex isolated from yeast could bind polyUb. Extracts from yeast cells expressing HA-tagged or untagged (control) β'-COP as the sole source of this subunit were incubated with anti-HA beads to recover COPI. The majority of β'-COP in yeast is assembled into the heptameric COPI complex and these methods have been used previously to purify the full complex (*Yip and Walz, 2011*). The beads carrying COPI bound K63-linked polyUb, but not K48-linked polyUb, with the same chain length specificity as the individual propeller domains (*Figure 3D–E*). In contrast, the beads incubated with control lysate with untagged COPI did not bind any of the polyUb chains. These data indicate the full-length β'-COP binds K63-polyUb and sug-gests the full COPI complex binds as well.

We then tested if COPI propeller domains are required for recycling of GFP-Snc1. Cells express-ing β'-COP harboring a deletion of the N-terminal propeller (Δ2–304) as the sole source of this sub-unit were viable but mislocalized GFP-Snc1 to Tlg1-marked compartments (*Figure 4A–C*). Deletion of the α-COP N-terminal propeller also disrupted Snc1 recycling, although not as severely (*Figure 4A–C*). The β'-COP N-terminal propeller contains a basic patch that binds to the C-terminus of cargos bearing specific variants of di-lysine sorting signals (e.g KxKxx-COO⁻), such as Emp47 (*Eugster et al., 2004*; *Schröder-Köhne et al., 1998*). Therefore, it was possible that the GFP-Snc1 recycling defect caused by β'-COP (Δ2–304) was a secondary effect of mislocalizing a subset of di-lysine cargos recognized preferentially by this propeller. Mutation of the di-lysine binding site (β'-COP RKR mutant: R15A K17A R59A) disrupts the di-lysine interaction and causes myc-Emp47 misloc-alization to the vacuole where it is degraded (*Eugster et al., 2004*). By contrast, the β'-COP RKR mutant localized GFP-Snc1 normally to the plasma membrane (*Figure 4A–C*). Therefore, the GFP-Snc1 recycling defect of β'-COP (Δ2–304) did not correlate with the loss of the di-lysine binding site. Importantly, these separation-of-function mutations show that the role of β'-COP at the Golgi in rec-ognizing certain di-lysine signals can be unlinked from its role in recycling GFP-Snc1.

## Replacement of the β'-COP N-terminal WD40 domain with unrelated Ub-binding domains restores Snc1 recycling

To test if recognition of Ub is the critical function of β'-COP in Snc1 recycling, we replaced the N-ter-minal propeller domain with three different Ub-binding domains. The first is a β-propeller Ub-bind-ing domain from Doa1 (UBD$_{Doa1}$) that has no significant sequence similarity to β'-COP, but is known to bind Ub without linkage or chain-length specificity (β'-COP UBD$_{Doa1}$) (*Pashkova et al., 2010*). The second is the Npl4 Zinc Finger (NZF) domain from Tab2, which binds specifically to K63-linked polyUb (*Sato et al., 2009*). The third is the K48-linkage specific UBA domain from Mud1 (*Trempe et al., 2005*). Strikingly, the UBD$_{Doa1}$ and NZF$_{Tab2}$ domains fully restored β'-COP function in GFP-Snc1 recycling (*Figure 4A–C*), but the UBA$_{Mud1}$ domain failed to support this trafficking path-way. Importantly, COPI Ub binding appears to be conserved because human β'-COP (1-303) bound K63-linked polyUb comparably to the orthologous yeast domain (*Figure 4—figure supplement 1*), and yeast cells expressing a chimeric yeast β'-COP with a human N-terminal propeller fully sup-ported GFP-Snc1 trafficking (*Figure 4A–D*).

β'-COP is encoded by the *SEC27* gene, which is essential for yeast viability. All of the β'-COP chi-meras and mutants described above supported the viability of yeast as the sole source of this sub-unit (*Figure 4E*). Therefore, all must be sufficiently well folded and functional to assemble into the heptameric complex. Interestingly, deletion of the β'-COP N-terminal propeller caused a slow

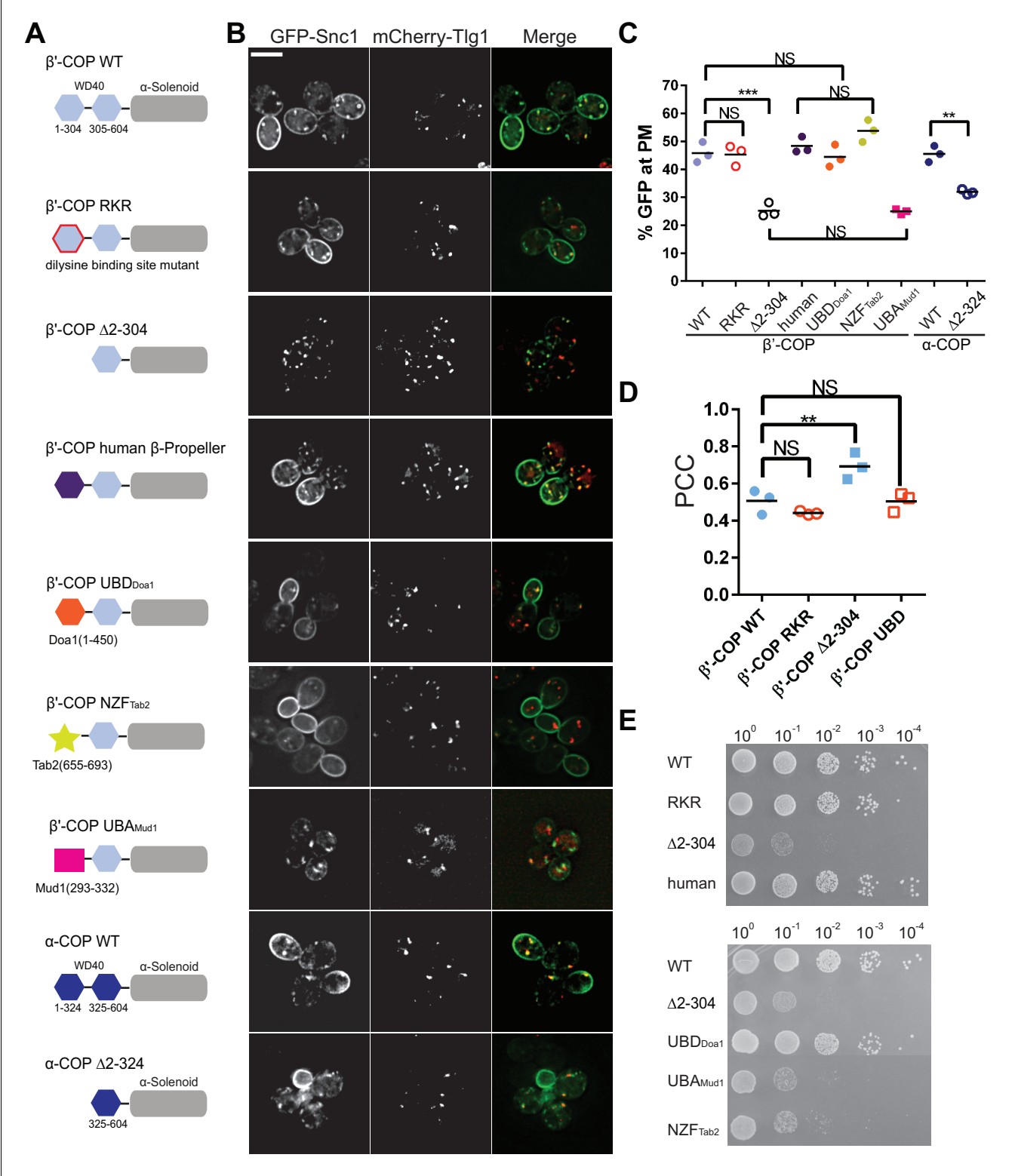

**Figure 4.** Ub binding by β'-COP is required to sort GFP-Snc1 to the plasma membrane. (**A and B**) Deletion of the N-terminal WD40 propeller from β'-COP (Δ2–304) disrupts recycling of GFP-Snc1 but mutation of residues within this propeller required for dilysine motif binding (RKR) have no effect. Replacement of the N-terminal propeller with a linkage independent ubiquitin-binding domain (UBD) from Doa1, a K63-specific Npl4 Zinc Finger (NZF) domain from Tab2, or the N-terminal propeller from human β'-COP restored Snc1 recycling. In contrast, the K48-linkage specific ubiquitin pathway associated (UBA) domain from Mud1 failed to restore GFP-Snc1 recycling. Deletion of N-terminal WD40 propeller of α-COP caused a partial Snc1

*Figure 4 continued on next page*

*Figure 4 continued*

recycling defect. Scale bar, 5 µm. (**C**) Quantification of GFP intensity at the plasma membrane. Each biological replicate includes at least 50 cells and individual biological replicates value and mean are shown. Statistical differences were determined using a one-way ANOVA on the means of the three biological replicates (***$p<0.001$; NS, $p>0.05$). (**D**) Quantification analysis of colocalization between GFP-Snc1 and mCherry-Tlg1 in WT and β'-COP mutant cells using Pearson correlation coefficient. Each replicate includes at least 20 cells and individual biological replicates value and mean were shown. (**E**) Serial dilution growth assay of β'-COP mutants. The β'-COP dilysine motif binding mutant (RKR) had no effect on growth, but deletion of the first propeller (Δ2–304) caused slow growth. Replacement of the first propeller with the UBD or the human N-terminal propeller, but not NZF or UBA domains, restored WT growth. One of four replicates is shown.

DOI: https://doi.org/10.7554/eLife.28342.010

The following source data and figure supplement are available for figure 4:

**Source data 1.** This spreadsheet contains the GFP intensity at the plasma membrane of COPI mutants used to generate the dot plots shown in *Figure 4C*, and the means of Pearson correlation coefficient data of COPI mutants used to generate the dot plots shown in *Figure 4D*.
DOI: https://doi.org/10.7554/eLife.28342.012
**Figure supplement 1.** GST tagged human β'-COP (1-303) preferentially binds long K63-linked (upper panel) but not K48-linked (middle panel) chains of Ub.
DOI: https://doi.org/10.7554/eLife.28342.011

growth phenotype that was fully rescued by its replacement with the general Ub-binding domain from Doa1 (UBD $_{Doa1}$), or the human β'-COP N-terminal propeller. The β'-COP (RKR) di-lysine binding mutant also supports WT growth. These results suggest that the slow growth phenotype caused by β'-COP (Δ2–304) was due to loss of Ub binding but not di-lysine binding. In contrast, β'-COP NZF$_{Tab2}$, which binds K63-linked Ub, failed to correct the growth defect even though it fully restored GFP-Snc1 recycling (*Figure 4A–E*). The β'-COP UBA$_{Mud1}$ chimera failed to correct the growth defect or Snc1 trafficking phenotypes exhibited by β'-COP (Δ2–304). The reason β'-COP NZF$_{Tab2}$ and β'-COP UBD$_{Doa1}$ influence growth differently is currently unclear, but suggest some COPI cargos may be modified with polyUb bearing linkages other than K63 or K48. These results also indicate that the growth and Snc1 trafficking defects can be uncoupled using different β'-COP variants.

It was possible that β'-COP (Δ2–304) destabilized the COPI coat and generally disrupted COPI function at the Golgi complex, thereby causing Snc1 recycling defects as a secondary consequence of perturbing the Golgi. Therefore, we examined the influence of these COPI variants on GFP-Rer1 cycling between the ER and Golgi complex. Rer1 is transported to the Golgi in COPII-coated vesicles and returned to the ER in COPI-coated vesicles, but displays a steady-state localization to early Golgi cisternae. Mutations that generally perturb COPI function, such as the temperature-sensitive *ret1-1* mutation in α-COP (*Letourneur et al., 1994*; *Sato et al., 2001*), mislocalize GFP-Rer1 to the vacuole (*Figure 5A,B*). Even at a permissive growth temperature of 27 °C, the *ret1-1* mutant displays significant mislocalization of Rer1-GFP to the vacuole (*Figure 5A,B*). By contrast, the β'-COP (Δ2–304), RKR and UBD$_{Doa1}$ mutants all localized GFP-Rer1 to the Golgi as efficiently as WT cells (*Figure 5A,B*). The β'-COP N-terminal di-lysine binding site has a specific role in sorting Emp47 within the Golgi. As previously reported, β'-COP (Δ2–304) and the RKR mutant mislocalizes Emp47 to the vacuole where it is degraded (*Eugster et al., 2004*). Replacement of the N-terminal propeller of β'-COP with the NZF$_{Tab1}$ or UBD$_{Doa1}$ domains predictably failed to stabilize Myc-Emp47 because these domains lack the di-lysine binding site (*Figure 5C*). We conclude the ability of β'-COP to bind ubiquitin is crucial for Snc1 recycling, but appears to have no role in the COPI-dependent trafficking of Rer1 or Emp47 at the Golgi complex. This collection of β'-COP fusion proteins provide an additional set of separation-of-function alleles that demonstrate the importance of the COPI-Ub interaction in vivo for GFP-Snc1 recycling.

## COPI localizes to Tlg1-marked membranes in budding yeast

Our data imply that COPI has a distinct function in Snc1 recycling that is independent of its role at early Golgi compartments, where most COPI is localized. To test whether COPI also localizes to compartments in the recycling pathway we quantified the colocalization of Cop1(α-COP)-mKate with GFP-Rer1 (early Golgi marker), GFP-Tlg1, and GFP-Sec7 (*Figure 6A,B*). While most COPI punctae colocalized with the early Golgi (61.3 ± 6.3%), we found 18.4 ± 3.6% of COPI co-localized with Tlg1, and only 2.5 ± 1.4% colocalized with Sec7 (*Figure 6A,B*). We also considered the possibility that Tlg1 was partially present in the early Golgi, which could provide an alternative explanation for the

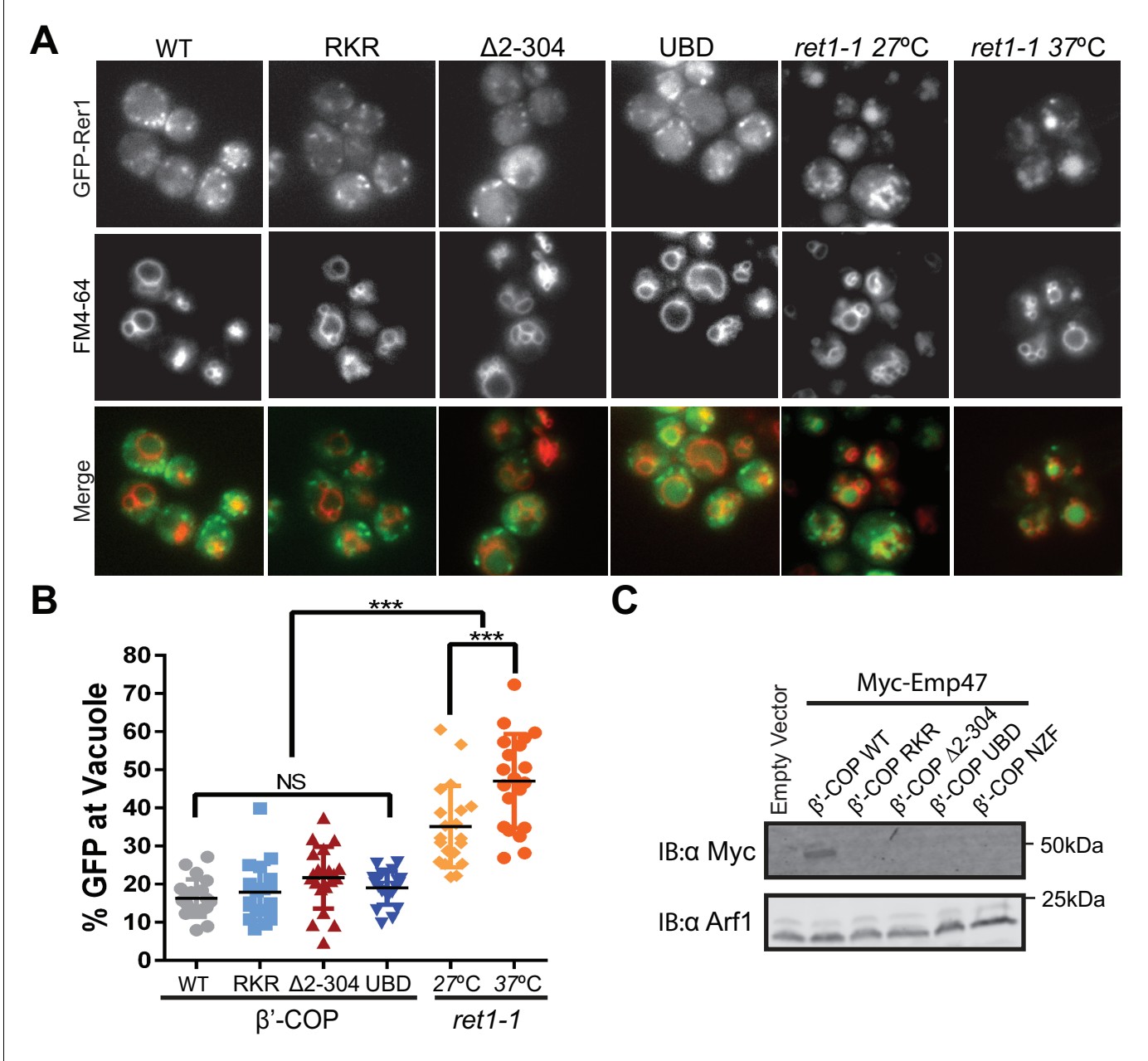

**Figure 5.** β'-COP interaction with ubiquitin has no role in the COPI-dependent trafficking of Rer1 or Emp47 at the Golgi complex. (**A**) Deletion of N-terminal WD40 propeller of β'-COP does not alter the retrograde trafficking of Rer1 from cis-Golgi to endoplasmic reticulum. The indicated β' COP mutant cells expressing GFP-Rer1 were labeled with 2 nM FM4-64 for 20 min at 30 ˚C then chased for 2 hr to label vacuole membranes. An α-COP temperature-sensitive mutant (*ret1-1*) expressing GFP-Rer1 was labeled with 2 nM FM4-64 at 27 ˚C for 20 min, then chased at 27 ˚C or 37 ˚C for 2 hr. (**B**) Quantification of GFP-Rer1 in the vacuole. The percentage values of GFP intensity in the vacuole for 20 cells were plotted. The mean and standard deviation were shown. Statistical differences were determined using a one-way ANOVA on the means of three biological replicates. (***p<0.001; NS, p>0.05). (**C**) Immunoblot showing that Myc-Emp47, an early Golgi COPI cargo, is missorted into the vacuole and degraded in strains expressing β'-COP RKR or Δ2–304, but stability is not restored when the N-terminal propeller is replaced with Ub binding domains. Immunoblot using anti-Arf1 is used as the loading control.

DOI: https://doi.org/10.7554/eLife.28342.013

The following source data is available for figure 5:

**Source data 1.** This spreadsheet contains the percentage of GFP intensity at the vacuole labeled with FM4-64 for individual β'-COPI mutants cells and *ret1-1* mutants cells used to generate the dot plots shown in *Figure 5B*.

DOI: https://doi.org/10.7554/eLife.28342.014

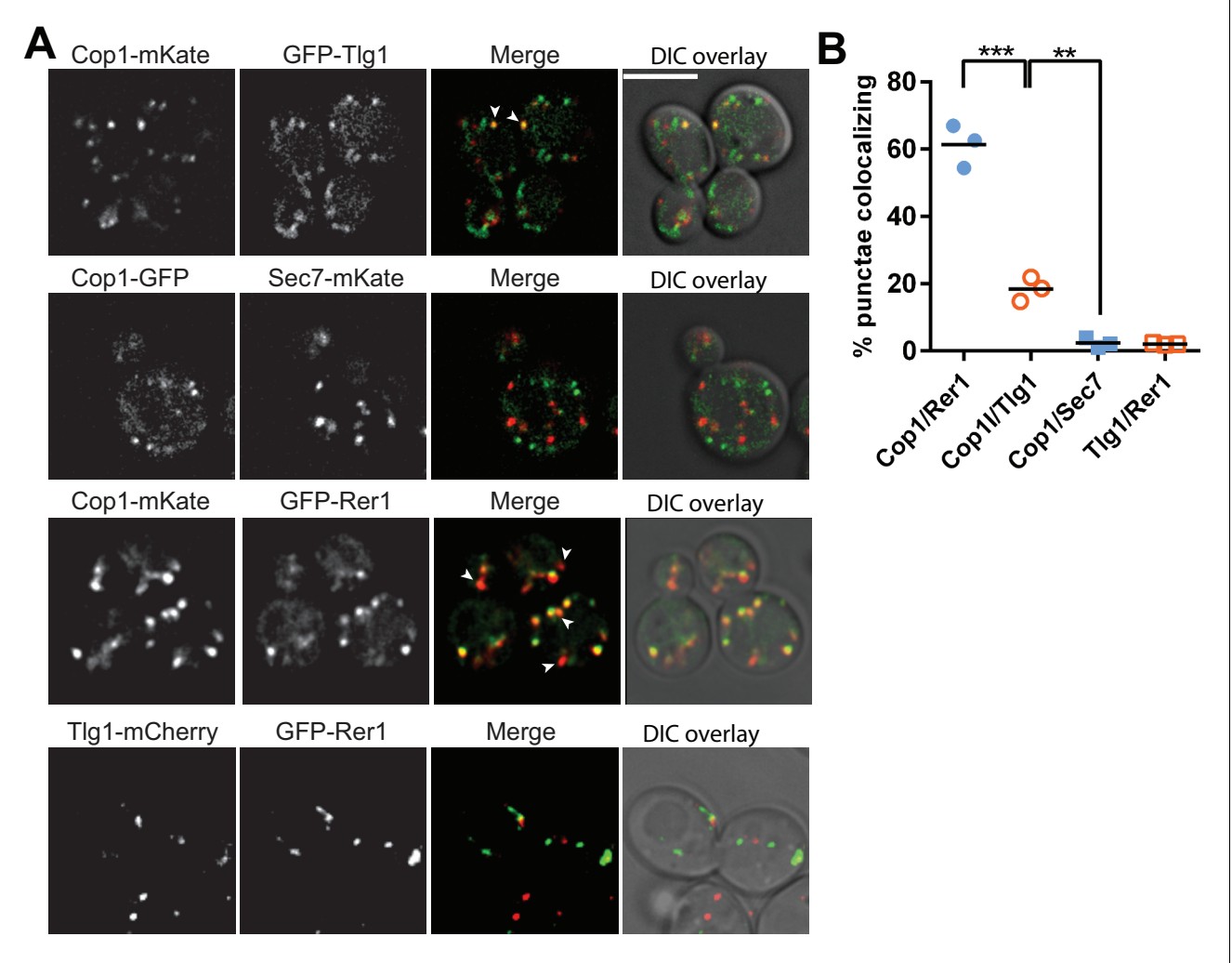

**Figure 6.** A small pool of COPI co-localizes with Tlg1 in yeast. (**A and B**) Co-localization of Cop1 (α-COP) with markers for the early endosome/TGN (Tlg1), early Golgi (Rer1), or TGN (Sec7). Tlg1 was also co-localized relative to Rer1 to make sure there was no significant overlap between these early Golgi and early endosome/TGN markers. Scale bar, 5 μm. Statistical differences were determined using a one-way ANOVA on the means of three biological replicates (***p<0.001; **p<0.01; NS, p>0.05).

DOI: https://doi.org/10.7554/eLife.28342.015

The following source data is available for figure 6:

**Source data 1.** This spreadsheet contains three means of percentage for Cop1 with Rer1, Tlg1 or Sec7 and Tlg1 with Sec7 data used to generate the dot plots shown in *Figure 6B*.

DOI: https://doi.org/10.7554/eLife.28342.016

co-localization with COPI. However, colocalization between mCherry-Tlg1 and GFP-Rer1 was negligible (2.4 ± 1.4%) (*Figure 6A,B*). Together, these observations suggest that COPI is recruited to compartments of the recycling pathway where it could play direct role in trafficking of ubiquitinated GFP-Snc1.

## Discussion

Here we report that the COPI vesicle coat protein recognizes K63-linked polyubiquitin and this interaction is necessary for recycling an exocytic v-SNARE Snc1 through the endocytic pathway back to the TGN. The significance of these observations to the protein trafficking field is that they (1) contrast with prevailing views on the role of ubiquitin in the endocytic pathway, (2) define a new and

unexpected sorting signal recognized by the COPI coat, and (3) define a novel mechanism for v-SNARE recycling.

## The multifaceted roles of Ub in protein trafficking

This study challenges the assumption that ubiquitination of membrane proteins in the secretory or endocytic pathways will target the modified protein solely to the vacuole for degradation. Ub is well known to be a sorting signal recognized by clathrin adaptor proteins at the TGN and plasma membrane, and by the ESCRT complexes in the endosomal system. The clathrin adaptor interactions initially sort the ubiquitinated cargo into the endosomal system, while the ESCRTs mediate sorting into intraluminal vesicles of multivesicular bodies for eventual delivery to vacuole (or lysosome) lumen where the cargo is typically degraded (*MacGurn et al., 2012*). In the yeast system, mono-ubiquitination of cargo appears to be sufficient to drive these sorting events (*Stringer and Piper, 2011*) and it is thought that Ub would have to be removed from cargo by a deubiquitinase in order to rescue the protein from vacuolar delivery. In contrast, we show a physiologically relevant example for how *addition* of a K63-linked polyUb chain onto a SNARE serves as a COPI-dependent sorting signal that diverts this cargo away from the endocytic pathway and mediates its retrieval from the endocytic pathway back to the TGN. A prior study had demonstrated that mutation of a conserved lysine in Snc1 (coincidentally K63) reduced its ubiquitination and perturbed recycling. However, it was possible that K63 in Snc1 was part of a more traditional lysine-based sorting signal in addition to being the primary target of ubiquitination. We rule out this possibility by the observation that fusion of a deubiquitinase (DUB) to GFP-Snc1 also disrupts its recycling without altering the Snc1 amino acid sequence. It is likely that DUB-GFP-Snc1 perturbs recycling by deubiquitinating Snc1 rather than the trafficking machinery because DUB fusion to Drs2 or Rcy1 did not cause the same Snc1 trafficking defect as observed with DUB-GFP-Snc1. However, we cannot rule out the possibility that deubiquitination of other components of the trafficking machinery contributes to this trafficking defect. Our observation that COPI binds directly to K63-linked poly-Ub chains and this interaction is necessary for Snc1 recycling strongly supports the idea that polyUb is the sorting signal in this pathway.

Our results provide a remarkable example of how the Ub code can be written and/or read in a spatiotemporally defined manner to control protein sorting decision points in the endocytic pathway. The Ub code is written by Ub ligases that determine the specific linkage and length of the Ub chains in competition with endogenous DUBs (*Komander et al., 2009*; *MacGurn et al., 2012*). The F-box protein Rcy1 has been implicated in Snc1 ubiquitination (*Chen et al., 2011*), but other SCF subunits are not required for Snc1 trafficking (*Galan et al., 2001*) and we find fusion of a DUB directly to Rcy1 does not perturb its function in this pathway relative to DUB*. However, DUB fusion with endosomal E3 ligases Pib1 and Tul1 does disrupt Snc1 recycling, as does combined deletion of the *PIB1* and *TUL1* genes. Thus, Pib1 and Tul1 are good candidates for the E3 ligases that modify Snc1. Mutations in both the membrane domain and membrane proximal cytosolic region of Snc1 perturb its recycling (*Lewis et al., 2000*), suggesting primary sorting information resides in these regions of the molecule. Tul1 is a candidate for recognizing a signal within the membrane domain because this E3 ligase is an integral membrane protein. However, it is also possible that a combination of a sorting receptor (unknown at this point) and ubiquitination drives Snc1 recycling. In addition, the regulation of ubiquitination/deubiquitination cycles for a substrate can be quite complex and more work is needed to clarify how the steady-state pools of Snc1-Ub are produced. With these caveats in mind, we speculate that Snc1 ubiquitination occurs at an early endosome population that lacks the ESCRT machinery so Snc1-Ub can be recycled by COPI rather than sorted into intraluminal vesicles. Conversely, it is possible that ESCRTs and COPI can compete for this cargo at the same compartment, but the length of the K63-linked polyUb chain on Snc1 determines whether COPI binds (long chains) or ESCRTs bind (mono or short chains). These events appear to be regulated because WT yeast harvested at a late phase of logarithmic growth display a substantial amount of GFP-Snc1 in the vacuole lumen (*MacDonald et al., 2015*). It will be interesting to determine if it is a regulated change in the trafficking machinery, or extent of Snc1 ubiquitination that elicits this switch in the Snc1 destination.

## K63-linked polyUb is a sorting signal that binds directly to COPI

COPI plays a major role in organelle biogenesis by helping establish the specific protein composition of the ER, Golgi complex, and endosomal membranes. This is accomplished by the recognition of sorting signals within the cytosolic tails of cargo proteins, such as the C-terminal di-lysine motif required for COPI-dependent Golgi to ER transport (*Cosson and Letourneur, 1994*; *Letourneur et al., 1994*). However, there is a paucity of information on other types of sorting signals potentially recognized by COPI. Here we show that the WD40 repeat domains of α- and β'-COP surprisingly bind to polyUb with specificity for the linkage type and length of the Ub chain. A chain of at least three K63-linked Ubs is required for productive interaction with the WD40 β-propeller domains. Indeed, it is tempting to speculate that the twin propeller domains of COPI evolved in order to bind long chains of Ub that could wrap around the two propeller domains. Moreover, we demonstrate the significance of the Ub interaction in vivo. Deletion of the ubiquitin-binding, N-terminal WD40 repeat domain of β'-COP disrupts endosome to TGN transport of Snc1, but this trafficking step can be restored when we replace the WD repeat domain with a 30-residue Tab2 NZF domain, which specifically binds K63-linked polyUb. By contrast, replacement of this β'-COP domain with a K48-linkage specific binding domain (Mud1 UBA domain) failed to restore Snc1 trafficking. Ub binding appears to be a conserved function for COPI as the human β'-COP N-terminal propeller also binds K63-linked ubiquitin and functionally replaces its yeast counterpart in the Snc1 recycling pathway. Importantly, ubiquitin binding by β'-COP has no discernable influence on COPI's role in Rer1 transport between the Golgi and ER, or the localization of Emp47. While it is possible that other COPI cargos may use ubiquitin signals early in the secretory pathway, we demonstrate a specific role for COPI in recycling through the endosomal system that is independent of its known functions in Golgi to ER transport.

## Exocytic v-SNARE recycling - will the circle be unbroken

The recycling of exocytic v-SNAREs is a multi-step process that begins with the fusion of these vesicles with the plasma membrane. For Snc1 and related mammalian v-SNAREs (e.g. VAMP2), the clathrin adaptor proteins AP180 and CALM recognize conserved Val and Met residues within the SNARE motif, which prevents re-association with the t-SNAREs and promotes endocytosis into clathrin coated vesicles (*Burston et al., 2009*; *Miller et al., 2011*). These exocytic v-SNAREs are not simply passive cargos in the subsequent steps as they mediate fusion of the endocytic vesicle with early endosomes (*Antonin et al., 2000*; *Holthuis et al., 1998b*; *McNew et al., 2000*). Snc1 is ubiquitinated on Lys49, Lys63 and Lys75 within the SNARE motif (*Swaney et al., 2013*) and these modifications likely inactivate Snc1 and prevent re-association with the endosomal t-SNAREs (Tlg1, Tlg2, and Vti1) after dissociation by Sec18/NSF. For example, monoubiquitination of Golgi SNAREs during mitosis is thought to inhibit their function as fusogens and facilitate Golgi fragmentation (*Huang et al., 2016*). We propose the polyUb chains on Snc1 also form the sorting signal that allows efficient departure from the Tlg1-marked compartment in COPI-coated vesicles. The ubiquitin signal appears to be a transient modification because only a small percentage of Snc1 is modified at steady-state, and it is likely that the ubiquitin is removed by an endogenous DUB to allow Snc1 to mediate fusion of recycling vesicles with their target.

Because the TGN and early endosomes in yeast are not well differentiated by existing markers, it is formally possible that the COPI and Ub influence we observe on Snc1 recycling reflects a defect in packaging Snc1 into exocytic vesicles rather than a defect in early endosome to TGN transport. However, most of the mutants that perturb Snc1 recycling (*rcy1Δ, drs2Δ, cdc50Δ, tlg1Δ, tlg2Δ)* have no measurable influence on the kinetics of secretion for other proteins (*Furuta et al., 2007*; *Holthuis et al., 1998a*; *Hua et al., 2002*; *Prescianotto-Baschong and Riezman, 2002*). Similarly, mutations that appear to fully inactivate COPI cause a cargo-selective defect in ER to Golgi transport, but cargos that do leave the ER normally are secreted with near normal kinetics (*Gaynor and Emr, 1997*). Moreover, a Snc1 mutation that disrupts its endocytosis is epistatic to mutations in the recycling machinery or the addition of the DUB (DUB-GFP-Snc1), which implies that the recycling defect caused by *rcy1Δ* (for example) or DUB fusion occurs after delivery to the plasma membrane and endocytosis, but prior to arrival back in the Golgi (*Figure 1A–B*) (*Chen et al., 2011*; *Mioka et al., 2014*). These observations argue against a role for COPI and the Rcy1/Drs2 machinery

in packaging Snc1 into exocytic vesicles at the TGN, and support a role for these factors in early endosome to TGN transport.

## Relationship of COPI to other factors mediating Snc1 recycling

A direct role for COPI in Snc1 recycling can illuminate the potential functions of the other trafficking components in this pathway. COPI is recruited to membranes by the small GTP binding protein Arf, which is regulated by multiple ArfGEFs and ArfGAPs (*Jackson and Casanova, 2000*). We have previously shown that the ArfGAP Gcs1 is specifically recruited to Tlg1-marked compartments by its ability to sense the curvature and charge imparted to the membrane by the Drs2 phosphatidylserine flippase (*Xu et al., 2013*). Gcs1 binds directly to Snc1 and COPI (*Robinson et al., 2006*; *Suckling et al., 2015*), and these interactions likely stabilize the COPI-Ub interactions to productively recruit Snc1 into the COPI-coated vesicles. Deletion of *GCS1* has a modest influence on Snc1 recycling relative to β'-COP (Δ2–304) or *drs2Δ* (*Xu et al., 2013*) and so we expect there are other effectors of Drs2 acting in the pathway and that the COPI interaction with Ub-Snc1 is primarily responsible for the sorting reaction. The F-box protein Rcy1 binds directly to a regulatory domain in the C-terminus of Drs2 (*Hanamatsu et al., 2014*), and binding of an ArfGEF (Gea2) to this Drs2 regulatory domain stimulates phosphatidylserine flippase activity (*Hsu et al., 2014*; *Natarajan et al., 2004*). Thus, the function of Rcy1 in this pathway may be activation of Drs2. The relationship of the Snx4/41/42 complex to the COPI-Gcs1-Drs2-Rcy1 network is less clear, but a recent study suggests this sorting nexin complex may be localized to a different endosome population and could represent a distinct pathway for retrieval of Snc1 (*Hettema et al., 2003*; *Ma et al., 2017*). Further work will be needed to determine precisely how these components work together with COPI to drive Snc1 transport from endosomes to the TGN.

In summary, our studies identify a new function for an old coat and define a specific trafficking function for COPI in recycling an exocytic SNARE. Further work is required to determine if mammalian exocytic SNAREs are recycled by this same mechanism and to identify other cargos that use a ubiquitin signal for sorting by COPI.

# Materials and methods

## Reagents

EZview Red ANTI-FLAG M2 Affinity Gel (F2426), EZview Red ANTI-HA Affinity Gel (E6779), 3xFLAG Peptide (F4799), HA Peptide (I2149), Glutathione−Agarose (G4510), N-Ethylmaleimide (S1638), Iodoacetamide (GERPN6302), aprotinin (A1153), pepstatin (P5318) and Phenylmethanesulfonyl fluoride (P7626) were purchased from Sigma-Aldrich (St Louis, MO). K48-, and K63-linked poly-ubiquitin chains (Ub2-7) were purchased from Boston Biochem (Cambridge, MA) and K63-linked tetra-ubiquitin was purchased from UBPBio (Aurora, CO). Protease inhibitor tablet (PI88665), Coomassie Brilliant Blue R-250 Dye (20278), and FM4-64 dye (T-3166) were purchased from ThermoFisher Scientific (San Jose, CA). ECL Prime Western Blotting Detection Reagent (RPN2236) was purchased from GE healthcare Life Sciences (Marlborough, MA).

## Antibodies

The rabbit anti-Arf1 (1:10,000) and rabbit anti-Drs2 (1:2000) antibodies have been described previously (*Chen et al., 1999*). Mouse anti-GFP (1C9A5, 1:2000) and mouse anti-Myc (9E10, 1:2000) antibodies were purchased from the Vanderbilt Antibody and Protein Repository (Nashville, TN). Mouse anti-FLAG M2 (Sigma-Aldrich Cat# F3165, RRID:AB_259529, 1:5000) and mouse anti-HA 12CA5 (Sigma-Aldrich Cat# 11583816001, RRID:AB_514505,1:2500) were purchased from Sigma-Aldrich. Mouse anti-Ubiquitin Ubi-1 antibody (Millipore Cat# MAB1510, RRID:AB_2180556, 1:1000) was purchased from EMD Millipore (Billerica, MA). Mouse anti-Ubiquitin VU1 (LifeSensors Cat# VU101, 1:1000) was purchased from LifeSensors (Malvern, PA). All secondary antibodies, including IRDye 680LT Goat anti-Mouse (LI-COR Biosciences Cat# 827–11080, RRID:AB_10795014, 1:20,000), IRDye 800CW Goat anti-Mouse (LI-COR Biosciences Cat# 827–08364, RRID:AB_10793856, 1:20,000), and IRDye 680LT Goat anti-Rabbit (LI-COR Biosciences Cat# 827–11081, RRID:AB_10795015, 1:20,000), were purchased from LI-COR Biosciences (Lincoln, NE).

## Strains and plasmids

Standard media and techniques for growing and transforming yeast were used. Epitope tagging of yeast genes was performed using a PCR toolbox (*Janke et al., 2004*). COPI mutant strains were constructed by plasmid shuffling (PXY51) on 5'-fluoro-orotic acid (5-FOA) plates. Plasmid constructions were performed using standard molecular manipulation. Mutations were introduced using a Q5 Site-Directed Mutagenesis Kit or Gibson Assembly Master Mix (New England BioLabs, Beverly, MA). *Supplementary file 1*. List of plasmids used in this study; *Supplementary file 2*. List of yeast strains used in this study.

DUB comes from herpes simplex virus UL36 and DUB* was constructed by point mutation C56S and deletion of ubiquitin binding β-hairpin (130-147) on DUB.

## Imaging and image analysis

To visualize GFP- or mCherry-tagged proteins, cells were grown to early-to-mid-logarithmic phase, harvested, and resuspended in imaging buffer (10 mM $Na_2PHO4$, 156 mM NaCl, 2 mM $KH_2PO4$, and 2% glucose). Cells were then mounted on glass slides and observed immediately at room temperature. Most images were acquired using a DeltaVision Elite Imaging System (GE Healthcare Life Sciences, Pittsburgh, PA) equipped with a 63 × objective lens followed by deconvolution using soft-WoRx software (GE Healthcare Life Science). All other images were acquired using an Axioplan microscope (Carl Zeiss,Thornwood, NY) equipped with a 63 × objective lens with an sCMOS camera (Zyla ANDOR, Belfast, United Kingdom) and Micro-Manager software. Overlay images were created using the merge channels function of ImageJ software (National Institutes of Health). GFP-Snc1 at the plasma membrane is quantified as previously described (*Hankins et al., 2015*). Briefly, concentric circles were drawn just inside and outside the plasma membrane using Image J to quantify the internal fluorescence and total fluorescence, respectively. The internal fluorescence was subtracted from the total to give the GFP intensity at the plasma membrane. At least 50 randomly chosen cells from three biological replicates (independently isolated strains with the same genotype) were used to calculate the mean and standard deviation. To quantify GFP-Snc1 colocalization with Tlg1, a Pearson's Correlation Coefficient (PCC) for the two markers in each cell (n = 3, 20 cells each) was calculated using the ImageJ plugin Just Another Colocalization Plugin with Costes Automatic Thresholding (*Bolte and Cordelières, 2006*). The percentage of COPI colocalization with the different organelle markers was calculated by counting how many COPI punctae (n = 3,>100 punctae each) colocalized with the markers.

## Purified recombinant proteins

GST-tagged recombinant proteins were expressed and purified as previously described (*Jackson et al., 2012*). Briefly, BL21(DE3)-pLysS (Agilent Technologies) *Escherichia coli* cells containing plasmids encoding each fusion protein were grown in 6 L of YT medium (16 g Tryptone, 10 g Yeast Extract and 5 g NaCl) containing 100 mg/ml ampicillin at 240 rpm at 37°C to an OD600 of 0.8. The expression was induced with 0.2 mM IPTG overnight at 22°C. Cells were harvested by centrifugation at 5,000 g for 10 min and stored at −80°C. Cells expressing β'-COP constructs were lysed in 20 mM Tris pH 7.4, 200 mM NaCl, 2 mM DTT, 2 µg/µl aprotinin, 0.7 µg/ml pepstatin. α-COP GST fusion proteins were purified in 20 mM Tris, pH 7.4, 500 mM NaCl, 2 mM DTT, 2 µg/µl aprotinin, 0.7 µg/ml pepstatin. Cells were lysed by a disruptor (Constant Systems Limited, Daventry, UK), and the lysates were centrifuged at 30,000 rpm for 1 hr. The supernatant was incubated with 5 ml of glutathione-agarose beads 1 hr at 4°C. The beads were washed in a column with 200 ml of washing buffer (20 mM Tris-HCl and 200 mM NaCl, pH 7.5), then eluted in 1 ml fractions with GST elution buffer (50 mM Tris-HCl and 20 mM reduced glutathione, pH 9.5). The protein was equilibrated to neutral buffer (20 mM Tris-HCl and 100 mM NaCl, pH 6.8) using dialysis. All proteins were further purified by gel filtration on a Superdex S200 preparative or analytical column (GE Healthcare Life Sciences, Pittsburgh, PA). Protein concentrations were measured by BCA assay (Sigma-Aldrich).

## In vitro binding assays

GST recombinant proteins were incubated with glutathione agarose beads in PBS at 25°C for 30 min. GST fusions on beads were then incubated with 10x molecular amount of $Ub_4$ at 25°C for 1 hr in incubation buffer (10 mM $Na_2PHO4$, 156 mM NaCl, 2 mM $KH_2PO4$, 0.1 mg/ml BSA, and 0.01%

Triton-X 100). Beads were then washed three times with wash buffer (10 mM $Na_2PHO4$, 156 mM NaCl, 2 mM $KH_2PO4$, and 0.01% Triton) and eluted with 50 mM reduced Glutathione in PBS on ice for 10 min. The elution was then mixed with SDS-Urea sample buffer at 60°C for 10 min.

For the ubiquitin chain binding, 0.5 μM of GST or GST tagged β-Prop was incubated with 9 μg of K63 or K48-linked ubiquitin chains in binding buffer (20 mM HEPES pH 7.5, 100 mM NaCl, 20% Glycerol, 0.1 % NP-40, 200 μg/ml BSA) at 4°C overnight. The GST beads were then added for 30 min. After three washes, the bound proteins were eluted with 50 mM reduced glutathione in PBS on ice for 10 min (*Sobhian et al., 2007*). The elution was then mixed with SDS-Urea sample buffer at 60°C for 10 min. The protein samples were analyzed by SDS–PAGE followed by immunoblotting using primary antibody and IRDye 680LT Goat anti-Mouse. The membrane was imaged with Licor Odyssey CLx (Licor, Lincoln, NE). The band intensities were quantified by Image Studio (Version 5.2). The relative binding was calculated as 100*(band signal intensity – corresponding GST lane)/input band intensity.

## NMR titration experiments

Uniformly enriched $^{15}$N-labeled ubiquitin prepared in 50 mM sodium phosphate pH 7.0, 1 mM DTT was diluted to 30 μM with 10% (v/v) $D_2O$. A sample of $^{15}$N-labeled ubiquitin with 10:1 molar excess of β'-COP residues 1–304 was prepared in the same way. Standard two-dimensional $^{15}$N-$^{1}$H HSQC spectra were collected at 25 °C on an 800 MHz Bruker Avance III spectrometer with a TCI triple resonance cryoprobe (Bruker BioSpin, Billerica, MA). Data were processed in Topspin 3.2 (Bruker BioSpin), with zero filling in the indirect dimension and squared sine bell apodization in both dimensions.

## Construction of HA tagged β'-COP

β'-COP was C-terminally tagged with 6xHA by integration of a PCR product amplified from pYM15 into the *SEC27* locus (*Janke et al., 2004*). Properly integrated clones were confirmed by genotyping PCR and immunoblot using anti-HA antibody.

## Purification of yeast coatomer for in vitro Ub binding assays

Affinity isolation of COPI was performed as previously described (*Yip and Walz, 2011*) with the following modifications. 2 L of wild type (BY4742) and C-terminal tagged 6xHA β'-COP (PXY57) yeast cells grown in YPD were pelleted when the OD600 reached ~4. After washing with cold water, the pellets were frozen. 5,000 OD of cells were resuspended in lysis buffer (10 mM Tris pH 7.4, 150 mM NaCl, 0.1% NP40, 2 mM EDTA, 50 mM NaF, 0.1 mM $Na_3VO_4$, 10 mM β-mercaptoethanol, 1 mM PMSF, and complete protease inhibitor tablet). Cells were broken using a Disruptor Genie (Scientific Industries, Bohemia, NY) at 4°C for 10 min at 3000 setting with 0.5 mm diameter of glass beads. The lysates were centrifuged at 13,000 rpm for 20 min at 4°C and the supernatant was incubated with anti-HA agarose beads for 1 hr at 4°C. The anti-HA agarose beads were washed with 1 ml of lysis buffer three times.

## Yeast coatomer binding assay

The anti-HA beads with bound coatomer were incubated with 4 μg of ubiquitin ladder mixtures in 500 μl of binding buffer (20 mM HEPES pH 7.5, 100 mM NaCl, 20% Glycerol, 0.1 % NP-40, 200 μg/ ml BSA) at 4°C for 2 hr. After the beads were washed three times, the specifically bound polyubiquitin was eluted from the beads by 3xHA peptide (100 μg/ml). The eluate was added to SDS-Urea sample buffer and heated at 60°C for 10 min. The protein samples were analyzed by SDS–PAGE followed by immunoblotting using primary anti-Ub antibody and anti–mouse IgG-HRP (1: 50,000 in TBST +5% non-fat milk). The membrane was developed by enhanced chemiluminescence (Amersham) and imaged with ImageQuant LAS 4000 (GE Healthcare Life Sciences, Pittsburgh, PA). The band intensity was quantified by ImageQuant TL (GE Life Sciences). The relative binding was calculated as 100* (pulldown band intensity/the input band intensity).

## HA-Snc1 immunoprecipitation

Wild-type yeast cells (BY4742) expressing empty vector (pRS416), lysineless Snc1 (pRS416-3HA-Snc1 8KR) and wildtype Snc1 (pRS416-3HA-Snc1) were inoculated followed by the subculture till $OD_{600}$

reaches to 0.6. ~ 100 OD of cells were collected and washed with cold double-distilled water. The cell pellets were treated with 0.2M NaOH at room temperature for 2 min. The re-pelleted cells were suspended with 200 μl of Urea-SDS buffer (50 mM Tris, pH 6.8, 8 M urea, 5% SDS, 10% glycerol, 10 mM N-ethylmaleimide, and 10 mM iodoacetamide) and heated at 70°C for 10 min. The cell lysates were diluted with 10-fold volumes of buffer (0.1 M Tris, pH 7.5, 0.4% Triton X-100, 10 mM N-ethyl-maleimide and 10 mM iodoacetamide) and incubated on ice for 10 min. After centrifugation at 13,000 rpm for 30 min, the supernatant was incubated with anti-HA agarose beads overnight while 50 μl of supernatant was taken as the loading control. Beads were subject to three 10 min washes with washing buffer (0.1 M Tris, pH 7.5, 0.4% Triton X-100). The HA tagged proteins were eluted with 25 μl of HA peptides (100 ng/μl in PBS) at 4°C for 30 min. The eluate was mixed with 2x SDS-Urea sample buffer (40 mM Tris-HCl, pH 6.8, 8 M urea, 0.1 mM EDTA, 1% β-mercaptoethanol, and 5% SDS) and was heated at 70°C for 10 min. The samples were run in 4–20% gradient SDS-PAGE and then transferred onto PVDF membrane. The PVDF membrane was blocked in Odyssey Blocking Buffer in TBS for 1 hr at room temperature then incubated with anti-HA antibodies (1:2500) in block-ing buffer overnight. After washing with TBST, the membrane is incubated with anti-mouse IgG-HRP (1: 50,000 in TBST +5% non-fat milk) for 1 hr at room temperature. The membrane was washed by TBST and then detected with ECL Prime Western Blotting (Amersham). The membrane was imaged with ChemiDoc MP (Bio-Rad, Hercules, CA).

## Enrichment of ubiquitinated proteins by Anti-FLAG immunoprecipitation

Immunoprecipitation was performed as described previously (*Stringer and Piper, 2011*) with the fol-lowing modifications. 50 OD of cells at mid-log phase were pelleted and resuspended in 0.2 M NaOH for 2 min. The cells were pelleted and resuspended in Urea-SDS buffer (50 mM Tris, pH 6.8, 8 M urea, 5% SDS, 10% glycerol, 10 mM N-ethylmaleimide, and 10 mM iodoacetamide) and boiled at 70°C for 10 min. The cell lysates were diluted with 10 volumes of 0.1 M Tris, pH 7.5, 0.4% Triton X-100, 10 mM N-ethylmaleimide and 10 mM iodoacetamide and placed on ice for 10 min. After cen-trifugation at 13,000 rpm for 30 min, the supernatant was incubated with anti-FLAG agarose beads overnight. Beads were washed three times in 0.1 M Tris, pH 7.5, 0.4% Triton X-100. Anti-FLAG beads were eluted with 20 μl 3xFLAG peptide (150 ng/μl in PBS) at 4°C for 30 min. The Supernatant mixed with 2x SDS-Urea sample buffer (40 mM Tris-HCl, pH 6.8, 8 M urea, 0.1 mM EDTA, 1% β-mer-captoethanol, and 5% SDS) was heated at 70°C for 10 min. The samples were then separated by 4–20% gradient SDS-PAGE followed by immunoblotting using the manufacturer's protocol (LI-COR Biosciences). PVDF membranes were scanned by an Odyssey CLx scanner and quantified using Image Studio Software (LI-COR Biosciences). To detect ubiquitinated proteins from yeast, the PVDF membrane was incubated anti-FLAG antibody (1:2500) followed by the incubation with anti-mouse IgG-HRP (1: 50,000 in TBST + 5% non-fat milk) for 1 hr at room temperature. After three washes in TBST, membranes were incubated with enhanced chemiluminescence (Amersham). The membrane was imaged with ImageQuant LAS 4000.

## Quantification of GFP-Rer1 in the vacuole

To label the vacuole of yeast cells, the cells expressing GFP-Rer1 were pulsed with 2 nM of FM4-64 at 30°C for 20 min. Then the cells were chased in YPD for 2 hr (*Vida and Emr, 1995*) before the images were acquired using an Axioplan microscope (Carl Zeiss) equipped with a 63 × objective lens with an sCMOS camera (Zyla ANDOR) and Micro-Manager software. Overlay images were cre-ated using the merge channels function of ImageJ software (National Institutes of Health). In ImageJ, the vacuole of a cell stained with FM4-64 was selected using the freehand draw tool and the same area was copied into the green (GFP) channel. The whole cell area was also defined using the free-hand draw tool and the GFP in the vacuole is defined as $Intensity_{vacuole}/Intensity_{whole\ cell}$. 20 cells were selected and quantified $\pm$ SD.

## Statistical analysis

Statistical differences were determined using a one-way ANOVA on the means of at least three inde-pendent experiments using GraphPad Prism (GraphPad Software Inc.). Probability values of less than

0.05, 0.01 and 0.001 were used to show statistically significant differences and are represented with *, ** or *** respectively.

## Acknowledgement

We thank Scott Emr (Cornell University), Natasha Pashkova (University of Iowa), Richard Chi and Chris Burd (Yale University), Thomas Mund and Hugh Pelham (MRC) and Daniel Finley (Harvard Medical School) for plasmids and yeast strains. We thank Jeffrey Gerst (Weizmann Institute of Science) for anti-Snc1 antibodies. These studies were supported by NIH Grants 1R01GM118452 (to TRG), 5R01GM058202 (to RCP), 1R35GM119525 (to LPJ) and 1R01GM118491 (to JAM). Lauren P Jackson is a Pew Scholar in the Biomedical Sciences, supported by the Pew Charitable Trusts.

## Additional information

### Funding

| Funder | Grant reference number | Author |
| --- | --- | --- |
| National Institutes of Health | 5R01GM118452 | Todd R Graham |
| Pew Charitable Trusts | | Lauren P Jackson |
| National Institutes of Health | 5R01GM058202 | Robert C Piper |
| National Institutes of Health | 1R35GM119525 | Lauren P Jackson |
| National Institutes of Health | 1R01GM118491 | Jason A MacGurn |

The funders had no role in study design, data collection and interpretation, or the decision to submit the work for publication.

### Author contributions

Peng Xu, Conceptualization, Formal analysis, Validation, Investigation, Visualization, Methodology, Writing—original draft, Writing—review and editing, Designed and performed the majority of the experiments and analyzed results; Hannah M Hankins, Validation, Investigation, Writing—review and editing, Analyzed image data and revised the paper; Chris MacDonald, Investigation, Methodology, Writing—review and editing, Designed and performed the E3 ligase experiments; Samuel J Erlinger, Investigation, Purified alpha-COP and revised the paper; Meredith N Frazier, Investigation, Performed the NMR experiments and revised paper; Nicholas S Diab, Validation, Writing—review and editing; Robert C Piper, Conceptualization, Supervision, Funding acquisition, Writing—review and editing, Designed the E3 ligase and NMR experiments. Critically revised the paper; Lauren P Jackson, Conceptualization, Resources, Funding acquisition, Visualization, Writing—review and editing, Purified all of the WD40 repeat domains. Created COPI diagram and critically revised the paper; Jason A MacGurn, Conceptualization, Resources, Supervision, Funding acquisition, Methodology, Writing—review and editing, Designed the experiments and provided sources like DeltaVision Microscope and antibodies. Critically revised the paper; Todd R Graham, Conceptualization, Supervision, Funding acquisition, Methodology, Writing—original draft, Project administration, Writing—review and editing, Supervised the whole project, proposed the ideas, critical check of the data and wrote/revised the paper

### Author ORCIDs

Peng Xu http://orcid.org/0000-0001-7103-3692
Samuel J Erlinger http://orcid.org/0000-0001-8347-2617
Lauren P Jackson https://orcid.org/0000-0002-3705-6126
Jason A MacGurn https://orcid.org/0000-0001-5063-259X
Todd R Graham http://orcid.org/0000-0002-3256-2126

### Decision letter and Author response

Decision letter https://doi.org/10.7554/eLife.28342.020

Author response https://doi.org/10.7554/eLife.28342.021

## Additional files

**Supplementary files**

• Supplementary file 1. List of plasmids used in this study.
DOI: https://doi.org/10.7554/eLife.28342.017

• Supplementary file 2. List of strains used in this study.
DOI: https://doi.org/10.7554/eLife.28342.018

• Transparent reporting form
DOI: https://doi.org/10.7554/eLife.28342.019

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
