## [Decision Letter]

Thank you for submitting your article "COPI mediates recycling of an exocytic SNARE from endosomes by recognition of a ubiquitin sorting signal" for consideration by *eLife*. Your article has been reviewed by three peer reviewers, and the evaluation has been overseen by a Reviewing Editor and Ivan Dikic as the Senior Editor. The following individuals involved in review of your submission have agreed to reveal their identity: Charles Barlowe (Reviewer #3).

The reviewers have discussed the reviews with one another and the Reviewing Editor has drafted this decision to help you prepare a revised submission.

Summary:

This is a very interesting paper reporting new functions for both the COPI coat and ubiquitination. Using several approaches, the authors convincingly show that the v-SNARE Snc1, which facilitates the fusion of secretory vesicles with the plasma membrane in yeast, is recycled back to the Golgi by COPI, making use of ubiquitin as a sorting signal. Up until now, ubiquitin on a membrane protein has been thought of as a signal for degradation, and COPI has been assumed to function before the trans-Golgi network (there were reports of COPI on endosomes in mammalian cells over 20 years ago, but these studies were never really followed up). v-SNAREs, COPI, and ubiquitination are all ancient proteins and pathways, so other eukaryotes may use similar recycling mechanisms.

Essential revisions:

All three reviewers were enthusiastic about both the quality of the work and the importance of the findings. They made a number of comments that will need to be addressed (see below), but most of the essential revisions require little or no experimental work, being more about clarity. Particularly important is the concern raised by Reviewer #1 about the identity of the compartment where COPI acts to sort Snc1. The authors use colocalisation with Tlg1 as evidence that the compartment is an early endosome, but as the reviewer points out, the TGN and early endosomes are closely linked, and Tlg1 cycles between both. The somewhat ambiguous nature of the "donor" department does not detract from the importance of the authors' conclusions that COPI can recognise ubiquitin as a sorting signal, and that this sorting step is away from the vacuole, rather than towards it as is generally assumed for ubiquitinated cargo proteins.

Reviewer #1:

This paper uses the yeast model to examine the sorting in the endocytic system of Snc1, the yeast ortholog of cellubrevin/VAMP3. In particular the authors investigate whether ubiquitination is required for the sorting of Snc1, and the find that it is and more strikingly, present evidence that the ubiquitin is recognised by a pool of the COPI vesicle coat that acts not in the Golgi, but in endosomes.

Overall, this is a potentially very interesting paper that includes some rather elegant experiments such attaching DUBs to Snc1, and also replacing a cargo binding domain of COPI with a known ubiquitin binding domain. As such I feel that it is potentially of the high standard of interest and technical quality expected for *eLife*, but there are one or two issues that need to be addressed, especially as the conclusion that COPI can act on early endosomes is slightly heretical and so needs robust proof, and at times more cautious wording.

a) The TGN and early endosomes are (EE) very closely linked, and in some ways they are more similar than assumed. For instance Tlg1 is used as an EE marker but it is in reality cycling between EE and the TGN and so should mark both. Likewise, the Tul1 Ub ligase component is evoked as contributing to ubiquitination of Snc1 in EEs, but previous data have suggested a role for Tul1 in the TGN. The authors need to carefully address how confident they are that the relevant location/activity of these proteins that they evoke really is the EE pool and not the TGN pool.

b) Results section, paragraph two: "Ubiquitination is required for EE to TGN transport" is an overstatement. In the absence of ubiquitination Snc1 does not go to vacuoles, and its accumulation in dots may simply mean that the rate-limiting step (or functional pathway) has changed – recycling from dots has been shown to be possible (Lewis et al., 2000). That ubiquitination accelerates retrieval is more accurate.

c) The Drs2-dub fusion control is OK, but is not conclusive. Suppose ubiquitination of a COP1 component were necessary, and Snc1 is in much closer association with COP1 than with Drs2? Again, caution required here.

d) Why does Pib1-DUB show an effect on Ub and sorting (Figure 2), but Pib1delta does not (Figure 2)?

e) Subsection “Replacement of the β’-COP N-terminal WD40 domain with unrelated Ub-binding domains restores Snc1 recycling”: The authors state that ubiquitin binding "has no role in the COPI-dependent trafficking of Rer1 or Emp47" but they actually can't say this for Emp47 as deleting the Ub binding site in COPI also deletes RxR binding site (and actually, Emp47 is also reported to be K63 ubiquitinated by Silva, Finley and Vogel, 2015).

f) Introduction section – the authors should cite Hettema et al., 2003 for Snx4 pathway – this is not in Lewis et al., 2000, and Hettema et al. long preceded Ma et al., 2017. This previous work shows clearly that Snx4 sorts Snc1, but the phenotype is different from the Rcy1/Ub pathway (Snc1 goes to the vacuole) and thus these are likely to be distinct pathways, with the Snx4 pathway being either in later endosomes or providing essential capacity to prevent traffic to the vacuole.

g) General point in Discussion: The authors data strongly suggests that ubiquitination is necessary, but it does not show that it is sufficient for sorting to the Golgi, and indeed they cite other interactions that could also be very important. This is relevant as it may help to explain why lots of things that get K63 ubiquitinated at the PM do not recycle to the Golgi – although the length of Ub chains may be part of the explanation, the authors do not present evidence that this is sufficient to discriminate. My guess is that other features, such as the TMD, play a major role. The main point the authors wish to make is that COPI mediates an EE-Golgi pathway, and that conclusion seems quite well supported.

h) It is not essential, but if the authors had any evidence that these findings are relevant to sorting of SNAREs in mammalian cells it would of course increase the impact of the work.

i) In the fluorescent micrographs the single channel images should be grey scale with only the merge in colour as this is considered best practice as it is easier to see, especially for the red-green colour blind (Figure 1, Figure 2, Figure 4, Figure 5, Figure 6).

Reviewer #2:

The manuscript by Xu et al. reports that in *S. cerevisiae* the vSNARE Snc1 is recycled from the endosomes to the TGN in a COPI dependent manner requiring the modification of SNC1 with K63-linked ubiquitin chains. Thus, the manuscript reports a number findings that will be of interests for scientist working on membrane trafficking and ubiquitination. In particular, the manuscript establishes a role for K63-linked ubiquitin chains as signal for the sorting of proteins from endosomes to the TGN. It also establishes a role of COPI in this sorting step. In my opinion, this manuscript is in principle a candidate for publication in *eLife*.

Major comments

1) Could the authors establish how widely the K63-linked ubiquitin chain and COPI mediated recycling from endosomes to the TGN might be? This could be done by testing by western blotting the co-precipitation of other proteins, which follow a similar trafficking route with the K63 specific TUBE, analogous to the experiment shown in Figure 2.

2) Along similar lines, the experiment shown in Figure 2 is not quite clear to me. I understand that the TUBEs precipitate all K48/K63-linked chains. Why is there almost no signal left in the K63 ubiquitin blot (right) for the K8R mutant? This would indicate that Snc1 is the only protein modified in this way. In addition, why does the ubiquitin positive signal run at such a high apparent molecular weight? How many ubiquitins would this correspond to? Further, I assume *Bkg refers to background. Where does this signal come from?

3) The authors should test the sorting of plasma membrane localized proteins in Pib1/Tul1 mutant cells.

4) The experiment shown in Figure 4—figure supplement 1 addressing the interaction of human β-COP with K63-linked ubiquitin chains is not convincing. Either the authors show an experiment of higher quality that clearly demonstrates the interaction above background or they should remove the figure from the manuscript.

5) The authors use Tlg1 as marker for endosomes throughout the study but employ the same protein as marker for endosomes and the TGN in Figure 6. Therefore, it is not clear to me if some of the increased co-localization of Snc1 and Tlg1 seen throughout the paper are due to reduced recycling from the endosomes of longer residence time in the TGN. Therefore, the increased localization of Snc1 to endosomes when its proposed ubiquitin-mediated traffic is interfered with should be backed up with a second endosomal marker.

Reviewer #3:

This manuscript by Xu et al., reports a significant finding in that COPI subunits recognize and return K63 polyubiquitylated Snc1 protein from early endosomes to the TGN. Biochemical experiments show that N-terminal WD40 propeller domains in β'-COPI and α-COPI bind directly to polyubiquitin chains and molecular genetic experiments show that N-terminal ubiquitin-binding domains in β'-COPI are required for Snc1 recycling in cells. Overall the experimental results support their concluding model and these findings explain longstanding questions regarding the coat machinery used for early endosome to TGN recycling. However, I suggest clarification of a few issues.

1) A strong prediction of the model is that when Snc1 is trapped in early endosomes, it should accumulate in the polyubiquitinated form. The authors mention in the discussion that at steady-state only a small fraction of Snc1 is modified. However, no comparison of Snc1 ubiquitination levels is shown between wild type and the endosome accumulated condition (e.g. ret1-1 or β'-COP delta2-304). If no changes were observed, explanation or further discussion of their model would be necessary.

2) In Figure 2, the middle panel in this experiment is not clear to me. Is this an experiment where lysates are immunoprecipitated with anti-HA and then blotted with anti-Ub? If these are anti-K63 TUBE precipitated samples then I would think anti-Ub would recognize many proteins and not just Ub-Snc1?

3) For Figure 3—figure supplement 1, there is no mention or interpretation in the text of the NMR experimental analysis shown.

4) A minor point, but for Figure 5 please clarify and include in the plasmid table that the construct used is ss-myc-EMP47, which places the myc-epitope just past the signal sequence cleavage site in Emp47.

5) In Figure 6, panel B indicates that 60% of Rer1 overlaps with COPI but in the panel A microscopy image there is little apparent overlap between COPI-mKate and GFP-Rer1? Is this a representative example or are the more optimal images to support this point.

[Editors' note: further revisions were requested prior to acceptance, as described below.]

Thank you for resubmitting your work entitled "COPI mediates recycling of an exocytic SNARE from endosomes by recognition of a ubiquitin sorting signal" for further consideration at *eLife*. Your revised article has been favorably evaluated by Ivan Dikic (Senior editor), a Reviewing editor, and three reviewers.

The manuscript has been improved but there are some remaining issues that need to be addressed before acceptance, as outlined below:

All three reviewers agree that the revised manuscript addresses nearly all of their comments on the original manuscript, but that there is still one outstanding concern: the statement that COPI is associated with early endosomes. The authors will need either to soften this claim considerably (including in the title), or to carry out additional work to support it. Reviewer 3 made a suggestion for a straightforward experiment, which both of the other reviewers felt would be a useful one that could yield important insights. The key points made by reviewers 1 and 3 are copied below:

"They provide quite good evidence that COPI is involved in Snc1 sorting, but I am less convinced by the evidence that COPI is on early endosomes. In particular they have not properly addressed the concern that their early endosome marker, Tlg1, is also in the TGN and trans-Golgi. They acknowledge that Tlg1 recycles between TGN and early endosomes, but then go on to argue that it is still a marker for early endosomes as it co-localizes in previous reports with FM4-64 and other markers of endocytosis. They also state that mutations in COPI resulting in Snc1 accumulating in a Tlg1 positive compartment can only be interpreted as accumulation in the early endosome, as if this compartment was the TGN it would mean COPI was binding Snc1 for exocytosis. However, this is not correct – it could be that some COPI is on the trans-Golgi for recycling within the Golgi stack and that mutations in COPI have an indirect effect on Golgi sorting and this delays Snc1's exit from the TGN. In this context it is disappointing that they do not cite and discuss the Fromme lab paper that uses live cell imaging to show that Tlg1 exits the Golgi slightly before Snc1, clearly implying that Tlg1 and Snc1 coexist in the same trans-Golgi compartments (McDonold and Fromme 2014)." (Reviewer 1)

"It does seem unclear where this sorting event occurs. There is also current debate about where/if sorting endosomes reside in yeast. In an attempt to comment constructively, I am not sure it is the authors' responsibility in this study to better define the yeast sorting endosome. But I do think they could better define where GFP-Snc1 accumulates in the β'-COP D2-304 condition. The authors argue that under this condition GFP-Snc1 should accumulate in a Tlg1+ and Sec7- compartment. Partial overlap with Tlg1 is shown, but I did not see an analysis of Sec7 co-localization. I agree with the concern that recycled GFP-Snc1 could be accumulating in a TGN compartment. It may be that this is a distinct region of the TGN and that Snc1 has to be recycled within the Golgi before it can rejoin exocytic cargo? From the experiment in Figure 6, the tools are available to measure overlap of GFP-Snc1 and Sec7-mKate in their mutant backgrounds. Regarding overlap of Sec7-RFP and GFP-Tlg1, Saimani et al. (2017) report 50% colocalization (PMID: 28521960)." (Reviewer 3)

---

## [Author Response]

Essential revisions:All three reviewers were enthusiastic about both the quality of the work and the importance of the findings. They made a number of comments that will need to be addressed (see below), but most of the essential revisions require little or no experimental work, being more about clarity. Particularly important is the concern raised by Reviewer #1 about the identity of the compartment where COPI acts to sort Snc1. The authors use colocalisation with Tlg1 as evidence that the compartment is an early endosome, but as the reviewer points out, the TGN and early endosomes are closely linked, and Tlg1 cycles between both. The somewhat ambiguous nature of the "donor" department does not detract from the importance of the authors' conclusions that COPI can recognise ubiquitin as a sorting signal, and that this sorting step is away from the vacuole, rather than towards it as is generally assumed for ubiquitinated cargo proteins.

We agree with this assessment and have made changes throughout the manuscript to better explain the relationship between these compartments, and to temper our conclusions where appropriate. We have added a new paragraph to the Introduction describing data in the literature implicating the Tlg1-marked punctae as early endosomes, as well as the overlap with TGN markers. We also better described the genetic rationale for why a vesicular transport step is thought to exist between an early endosomal compartment and TGN in budding yeast. We have tried to avoid making definitive statements about the nature of these compartments in recognition of the inherent ambiguity in giving a static label to highly dynamic organelles. In many places in the Results we’ve replaced “early endosome (EE)” with “puncta”, and a “defect in EE to TGN transport” with a “defect in recycling”. We also revisited the issue in the Discussion to explain why available data better support a role for COPI in EE to TGN transport than transport from the TGN, while acknowledging more than one interpretation is possible. Specific changes to the text are described below in response to the reviewers’ comments

Reviewer #1:This paper uses the yeast model to examine the sorting in the endocytic system of Snc1, the yeast ortholog of cellubrevin/VAMP3. In particular the authors investigate whether ubiquitination is required for the sorting of Snc1, and the find that it is and more strikingly, present evidence that the ubiquitin is recognised by a pool of the COPI vesicle coat that acts not in the Golgi, but in endosomes.Overall, this is a potentially very interesting paper that includes some rather elegant experiments such attaching DUBs to Snc1, and also replacing a cargo binding domain of COPI with a known ubiquitin binding domain. As such I feel that it is potentially of the high standard of interest and technical quality expected for eLife, but there are one or two issues that need to be addressed, especially as the conclusion that COPI can act on early endosomes is slightly heretical and so needs robust proof, and at times more cautious wording.a) The TGN and early endosomes are (EE) very closely linked, and in some ways they are more similar than assumed. For instance Tlg1 is used as an EE marker but it is in reality cycling between EE and the TGN and so should mark both. Likewise, the Tul1 Ub ligase component is evoked as contributing to ubiquitination of Snc1 in EEs, but previous data have suggested a role for Tul1 in the TGN. The authors need to carefully address how confident they are that the relevant location/activity of these proteins that they evoke really is the EE pool and not the TGN pool.

As described above, we have added a paragraph to the Introduction to better describe the relationship between the TGN and endosomal compartments in yeast, and why a vesicular transport step is thought to occur between functionally distinct early endosomes and TGN cisternae in yeast. We agree that there is not much of a distinction between these compartments as typically defined. However, the alternative interpretation of our data – that Ub and COPI mediate packaging of Snc1 into exocytic vesicles at the TGN for delivery to the plasma membrane – is poorly supported by the available data and is probably more heretical to propose. Importantly, the kinetics of secretion for rcy1∆, drs2∆, cdc50∆, tlg1∆, tlg1∆ tlg2∆ and various COPI mutants has been carefully analyzed in multiple labs with no indication for a delay in Golgi to plasma membrane transport for many different cargos. It is difficult to imagine that Snc1 is specifically retained in the TGN in these mutants while other cargos are not. The epistasis experiments with the endocytosis defective form of Snc1 (Figure 1, and previously performed with rcy1∆, drs2∆ and tlg1∆ mutants) supports the conclusion that endocytosis of Snc1 is required for accumulation of GFP-Snc1 in the internal, Tlg1^+^ punctae and that a defect in TGN to PM transport is unlikely to account for trafficking defect.

In addition, there is substantial functional evidence to describe the Tlg1^+^ organelle as an early endosome from both light and electron microscopy based kinetic experiments (with pulses of FM4-64 or positively charged gold particles endocytosed from the plasma membrane). These compartments are probably deficient for Sec7 (Tlg1^+^Sec7^-^) although it is challenging to follow 3 markers (endocytic tracer, Tlg1 and Sec7/Kex2) simultaneously in the same cell. I am unaware of any published data showing newly synthesized secretory and/or vacuolar proteins pass through Tlg1^+^ punctae prior to arrival at their destination (as would be expected if all Tlg1^+^ puncta were equal to the TGN). We believe the available evidence strongly supports the “early endosome” or “recycling endosome” description of this compartment, but recognize that more caution in interpretation is warranted. In addition to the new paragraph in the Introduction, we have added a paragraph in the Discussion that starts “Because the TGN and early endosomes in yeast are not well differentiated by existing markers, it is formally possible that the COPI and Ub influence we observe on Snc1 recycling reflects a defect in packaging Snc1 into exocytic vesicles rather than a defect in EE to TGN transport.”. We go on to more fully describe the reasons that this is unlikely. We also revised the text in several places – for example, “However, we cannot exclude the possibility that some punctae containing COPI and Tlg1 are TGN cisternae”. The Discussion now reads “Unfortunately, there are no markers described in the literature that uniquely identify the early/recycling endosome in yeast, but this Tlg1^+^ Sec7^-^ compartment is functionally and genetically implicated as the recycling endosome.”

b) Results section, paragraph two: "Ubiquitination is required for EE to TGN transport" is an overstatement. In the absence of ubiquitination Snc1 does not go to vacuoles, and its accumulation in dots may simply mean that the rate-limiting step (or functional pathway) has changed – recycling from dots has been shown to be possible (Lewis et al., 2000). That ubiquitination accelerates retrieval is more accurate.

We have changed the text to “The endocytosis-defective variants (e.g. DUB-GFP-Snc1-PM) accumulated at the plasma membrane (Figure 1), strongly suggesting that the DUB delayed early endosome to TGN trafficking rather than TGN to plasma membrane transport when attached to WT Snc1.”

c) The Drs2-dub fusion control is OK, but is not conclusive. Suppose ubiquitination of a COP1 component were necessary, and Snc1 is in much closer association with COP1 than with Drs2? Again, caution required here.

We have added this sentence to the Discussion: “It is likely that DUB-GFP-Snc1 perturbs recycling by deubiquitinating Snc1 rather than the trafficking machinery because DUB fusion to Drs2 or Rcy1 did not cause the same Snc1 trafficking defect as observed with DUB-GFP-Snc1. However, we cannot rule out the possibility that deubiquitination of other components of the trafficking machinery contributes to this trafficking defect.”

d) Why does Pib1-DUB show an effect on Ub and sorting (Figure 2), but Pib1delta does not (Figure 2)?

It is having a dominant phenotype as the strain also contains WT Pib1. We assume that Pib1-Dub has access to the substrate and can strip off the ubiquitin regardless of whether it is added by Pib1, Tul1 or any other Ub ligase. Genetically, PIB1 and TUL1 are acting redundantly in supporting GFP-Snc1 trafficking. The pib1∆ strain does not mislocalize GFP-Snc1 presumably because Tul1 can ubiquitinate it.

e) Subsection “Replacement of the β’-COP N-terminal WD40 domain with unrelated Ub-binding domains restores Snc1 recycling”: The authors state that ubiquitin binding "has no role in the COPI-dependent trafficking of Rer1 or Emp47" but they actually can't say this for Emp47 as deleting the Ub binding site in COPI also deletes RxR binding site (and actually, Emp47 is also reported to be K63 ubiquitinated by Silva et al.).

Emp47 does have an RxR motif adjacent to its membrane domain, but RxR does not appear to contribute to the COPI-dependent sorting of this protein. Mutation of one or both of the lysines in the KxKxx motif causes substantial mislocalization of Emp47 and so the protein relies strongly on the interaction of the dilysine motif with R15K17R59 (Schroder et al., 1995 JCB; Jackson et al., 2012). These observations and our data showing the Doa1 and NZF β’COP fusion proteins do not restore Emp47 stability strongly suggests that sorting of Emp47 does not rely on a COPIUb interaction. However, it is possible that such an interaction important for Emp47 trafficking is missed by our assays and so we revised this sentence to “…. but appears to have no role….”.

f) Introduction section – the authors should cite Hettema et al., 2003 for Snx4 pathway – this is not in Lewis et al., 2000, and Hettema et al. long preceded Ma et al., 2017. This previous work shows clearly that Snx4 sorts Snc1, but the phenotype is different from the Rcy1/Ub pathway (Snc1 goes to the vacuole) and thus these are likely to be distinct pathways, with the Snx4 pathway being either in later endosomes or providing essential capacity to prevent traffic to the vacuole.

Thank you for catching this mistake! The appropriate reference is included in the Introduction and references.

g) General point in Discussion: The authors data strongly suggests that ubiquitination is necessary, but it does not show that it is sufficient for sorting to the Golgi, and indeed they cite other interactions that could also be very important. This is relevant as it may help to explain why lots of things that get K63 ubiquitinated at the PM do not recycle to the Golgi – although the length of Ub chains may be part of the explanation, the authors do not present evidence that this is sufficient to discriminate. My guess is that other features, such as the TMD, play a major role. The main point the authors wish to make is that COPI mediates an EE-Golgi pathway, and that conclusion seems quite well supported.

We have added the following two sentences of the Discussion. “Mutations in both the membrane domain and membrane proximal region of Snc1 perturb its recycling suggesting primary sorting information resides in these regions of the molecule. Tul1 is a candidate for recognizing a signal within the membrane domain because this E3 ligase is an integral membrane protein. However, it is also possible that a combination of sorting receptor (unknown at this point) and ubiquitination drive Snc1 recycling.”

h) It is not essential, but if the authors had any evidence that these findings are relevant to sorting of SNAREs in mammalian cells it would of course increase the impact of the work.

We have just started working on this and hope that it will be a good follow up to the current manuscript.

i) In the fluorescent micrographs the single channel images should be grey scale with only the merge in colour as this is considered best practice as it is easier to see, especially for the red-green colour blind (Figure 1, Figure 2, Figure 4, Figure 5, Figure 6).

Thank you for this suggestion. We changed all the red and green fluorescent channel into grayscale in Figure 1, Figure 2, Figure 4, Figure 5 and Figure 6.

Reviewer #2:The manuscript by Xu et al. reports that in S. cerevisiae the vSNARE Snc1 is recycled from the endosomes to the TGN in a COPI dependent manner requiring the modification of SNC1 with K63-linked ubiquitin chains. Thus, the manuscript reports a number findings that will be of interests for scientist working on membrane trafficking and ubiquitination. In particular, the manuscript establishes a role for K63-linked ubiquitin chains as signal for the sorting of proteins from endosomes to the TGN. It also establishes a role of COPI in this sorting step. In my opinion, this manuscript is in principle a candidate for publication in eLife.Major comments1) Could the authors establish how widely the K63-linked ubiquitin chain and COPI mediated recycling from endosomes to the TGN might be? This could be done by testing by western blotting the co-precipitation of other proteins, which follow a similar trafficking route with the K63 specific TUBE, analogous to the experiment shown in Figure 2.

We have tested 9 plasma membrane proteins that were reported to be underubiquitinated (like Snc1) in yeast expressing only the K63R ubiquitin mutant. We observed a minor mislocalization phenotype for a few of these (Ena1, for example) in the β’ COP ∆2-304 mutant, but the phenotypes weren’t robust enough to give statistically significant differences from the WT controls. There may be multiple pathways operating in parallel to deliver proteins from the endosomal pathway to the Golgi (retromer, Snx4/41, Snx4/42, AP1/clathrin, Rcy1/Drs2/COPI) and Snc1 appears to be unusual in having such as strong dependence on the Rcy1/Drs2/COPI route. We are making a number of double mutant combinations to better tease out redundancies in these systems, but we feel this is moving us beyond the scope of the current work. We’ve recently started testing other SNARES and found a portion of GFP-Sec22 is mislocalized to the vacuole in the β’ COP ∆2-304 mutant, but is not mislocalized in the β’ COP-Doa1 strain. However, fully testing the other SNAREs and performing all the appropriate controls for Sec22 is a substantial undertaking that we feel is beyond the scope of the current work. Importantly, these observations do not alter the conclusions of this manuscript. However, we did revise the following sentence in the Discussion to raise the possibility that COPIUb interactions could also drive sorting events earlier in the secretory pathway. “While it is possible that other COPI cargos may use ubiquitin signals early in the secretory pathway, we demonstrate a specific role for COPI in recycling through the endosomal system that is independent of its known functions in Golgi to ER transport.”

2) Along similar lines, the experiment shown in Figure 2 is not quite clear to me. I understand that the TUBEs precipitate all K48/K63-linked chains. Why is there almost no signal left in the K63 ubiquitin blot (right) for the K8R mutant? This would indicate that Snc1 is the only protein modified in this way. In addition, why does the ubiquitin positive signal run at such a high apparent molecular weight? How many ubiquitins would this correspond to? Further, I assume *Bkg refers to background. Where does this signal come from?

This experiment has been a real challenge for us because it requires preparation of lysates in nondenaturing conditions and we’ve struggled to preserve Snc1Ub against the endogenous DUBs. We looked back through our older data and each experiment had an issue with at least one of the controls (reviewer 3 also pointed out problems with this experiment). We tried to repeat the experiment and retained very little polyubiquitinated Snc1 to analyze and the signal to noise was lousy. While the Figure 2 data we have supported the original conclusion (as do many other experiments in the manuscript), we felt the quality of the data was inadequate to stand on its own merits and did not meet the reviewers (or our own) expectations. Therefore, we removed the 2B panel from the figure and revised the text appropriately.

As a control for the K63-TUBE experiments, we prepared lysates under denaturing conditions and ended up with nicer immunoblots showing the polyubiquitinated forms of Snc1 than we originally submitted. We are now presenting one of these blots as Figure 2 and the original Figure 2 is now Figure 2.

3) The authors should test the sorting of plasma membrane localized proteins in Pib1/Tul1 mutant cells.

We have examined two membrane proteins, Ina1 and Tat1 in the pib1∆ tul1∆ strain and did not see a difference in the distribution between the plasma membrane and internal organelles. This result in now included as Figure 2—figure supplement 2 and described in subsection “Snc1 is extensively modified with polyUb chains”

4) The experiment shown in Figure 4—figure supplement 1 addressing the interaction of human β-COP with K63-linked ubiquitin chains is not convincing. Either the authors show an experiment of higher quality that clearly demonstrates the interaction above background or they should remove the figure from the manuscript.

The human propeller domain fused to GST does not express nearly as well in *E. coli* as the yeast counterpart. However, we felt this was an important experiment and wanted to retain the result in the manuscript. Therefore, we repeated the experiment and did a better job of purifying the human GST-β’COP(1-303) fusion. While the signal over background is not as strong as the yeast fusion, the new pull-down experiments (revised Figure 4—figure supplement 1) are more convincing to show a binding preference for long K63linked PolyUb chains relative to our original figure.

5) The authors use Tlg1 as marker for endosomes throughout the study but employ the same protein as marker for endosomes and the TGN in Figure 6. Therefore, it is not clear to me if some of the increased co-localization of Snc1 and Tlg1 seen throughout the paper are due to reduced recycling from the endosomes of longer residence time in the TGN. Therefore, the increased localization of Snc1 to endosomes when its proposed ubiquitin-mediated traffic is interfered with should be backed up with a second endosomal marker.

Unfortunately, we unaware of a better marker for the early endosome that we could use. FM4-64 is often used, but this follows the recycling pathway and labels Sec7 positive compartment as well as the Tlg1 positive compartment. To distinguish the arrival time of FM4-64 in these compartments is a difficult kinetic imaging experiment to do well. We have tried to express BFP fusion proteins to perform three-color analyses (FM464, Tlg1 and Sec7) but the fluorescence was too weak for us to detect. As detailed above, significant revisions to the text have been made reflect the complications in unambiguously defining the donor compartment. For example, “However, we cannot exclude the possibility that some punctae containing COPI and Tlg1 are TGN cisternae”. The Discussion now reads “Unfortunately, there are no markers described in the literature that uniquely identify the early/recycling endosome in yeast, but this Tlg1^+^ Sec7^-^ compartment is functionally and genetically implicated as the recycling endosome.”

Reviewer #3:This manuscript by Xu et al., reports a significant finding in that COPI subunits recognize and return K63 polyubiquitylated Snc1 protein from early endosomes to the TGN. Biochemical experiments show that N-terminal WD40 propeller domains in β'-COPI and α-COPI bind directly to polyubiquitin chains and molecular genetic experiments show that N-terminal ubiquitin-binding domains in β'-COPI are required for Snc1 recycling in cells. Overall the experimental results support their concluding model and these findings explain longstanding questions regarding the coat machinery used for early endosome to TGN recycling. However, I suggest clarification of a few issues.1) A strong prediction of the model is that when Snc1 is trapped in early endosomes, it should accumulate in the polyubiquitinated form. The authors mention in the discussion that at steady-state only a small fraction of Snc1 is modified. However, no comparison of Snc1 ubiquitination levels is shown between wild type and the endosome accumulated condition (e.g. ret1-1 or β'-COP delta2-304). If no changes were observed, explanation or further discussion of their model would be necessary.

We tried this experiment and initially observed a bit more Snc1-Ub in the ∆2304 mutant relative to WT, but on the second attempt observed about the same amount in both strains. At this point, the data are not robust enough to make a definitive claim that there is no difference between WT and mutant, but we are not seeing a substantial accumulation of Snc1-Ub when its recycling is perturbed. This result is hard to interpret until we better understand the ubiquitination/deubiquitination cycle for Snc1 and how it is coupled to the trafficking events in more detail. For example, it is possible that the interaction between COPI and Ub helps protect Snc1-Ub from DUBs and loss of this protective interaction offsets the expected increase in ubiquitination due to longer retention in the donor compartment. In response, we’ve revised the Discussion as follows. “In addition, the regulation of ubiquitination/deubiquitination cycles for a substrate can be quite complex and more work is needed to clarify how the steady-state pools of Snc1-Ub are produced. With these caveats in mind, we speculate that Snc1 ubiquitination occurs at an early endosome population that lacks the ESCRT machinery so Snc1-Ub can be recycled by COPI rather than sorted into intraluminal vesicles.”

2) In Figure 2, the middle panel in this experiment is not clear to me. Is this an experiment where lysates are immunoprecipitated with anti-HA and then blotted with anti-Ub? If these are anti-K63 TUBE precipitated samples then I would think anti-Ub would recognize many proteins and not just Ub-Snc1?

Please see the response to point 2 from reviewer 2 above as we have removed this experiment from the manuscript. The lysates were precipitated with the K48 or K63 TUBE reagents and then probed for either Snc1 (HA) or Ub. We are not sure why the strong bands showed up in the original middle panel. Another concern was the K63 polyUb 8KR control sample in the middle panel showed a poor recovery of K63-linked polyUb relative to the other samples, casting doubt on the lack of signal in this sample in the original top panel. For these reasons, and those cited above, we removed Figure 2.

3) For Figure 3, supplement 1, there is no mention or interpretation in the text of the NMR experimental analysis shown.

We added this sentence on in subsection “COPI binds K63-linked polyUb directly and this interaction is required for GFP-Snc1 recycling”: “For the latter experiment, we incubated ^15^N-ubiquitin with or without a 10-fold molar excess of β’-COP (1-304) and observed no difference in the HSQC NMR spectra, indicating no measurable interaction under these condition (Figure 3—figure supplement 1).”

4) A minor point, but for Figure 5 please clarify and include in the plasmid table that the construct used is ss-myc-EMP47, which places the myc-epitope just past the signal sequence cleavage site in Emp47.

Clarified as requested.

5) In Figure 6, panel B indicates that 60% of Rer1 overlaps with COPI but in the panel A microscopy image there is little apparent overlap between COPI-mKate and GFP-Rer1? Is this a representative example or are the more optimal images to support this point.

We replaced the original images with images that are more representative of the population average.

[Editors' note: further revisions were requested prior to acceptance, as described below.]

All three reviewers agree that the revised manuscript addresses nearly all of their comments on the original manuscript, but that there is still one outstanding concern: the statement that COPI is associated with early endosomes. The authors will need either to soften this claim considerably (including in the title), or to carry out additional work to support it. Reviewer 3 made a suggestion for a straightforward experiment, which both of the other reviewers felt would be a useful one that could yield important insights. The key points made by reviewers 1 and 3 are copied below:"They provide quite good evidence that COPI is involved in Snc1 sorting, but I am less convinced by the evidence that COPI is on early endosomes. In particular they have not properly addressed the concern that their early endosome marker, Tlg1, is also in the TGN and trans-Golgi. They acknowledge that Tlg1 recycles between TGN and early endosomes, but then go on to argue that it is still a marker for early endosomes as it co-localizes in previous reports with FM4-64 and other markers of endocytosis. They also state that mutations in COPI resulting in Snc1 accumulating in a Tlg1 positive compartment can only be interpreted as accumulation in the early endosome, as if this compartment was the TGN it would mean COPI was binding Snc1 for exocytosis. However, this is not correct – it could be that some COPI is on the trans-Golgi for recycling within the Golgi stack and that mutations in COPI have an indirect effect on Golgi sorting and this delays Snc1's exit from the TGN. In this context it is disappointing that they do not cite and discuss the Fromme lab paper that uses live cell imaging to show that Tlg1 exits the Golgi slightly before Snc1, clearly implying that Tlg1 and Snc1 coexist in the same trans-Golgi compartments (McDonold and Fromme 2014)." (Reviewer 1)"It does seem unclear where this sorting event occurs. There is also current debate about where/if sorting endosomes reside in yeast. In an attempt to comment constructively, I am not sure it is the authors' responsibility in this study to better define the yeast sorting endosome. But I do think they could better define where GFP-Snc1 accumulates in the β'-COP D2-304 condition. The authors argue that under this condition GFP-Snc1 should accumulate in a Tlg1+ and Sec7- compartment. Partial overlap with Tlg1 is shown, but I did not see an analysis of Sec7 co-localization. I agree with the concern that recycled GFP-Snc1 could be accumulating in a TGN compartment. It may be that this is a distinct region of the TGN and that Snc1 has to be recycled within the Golgi before it can rejoin exocytic cargo? From the experiment in Figure 6, the tools are available to measure overlap of GFP-Snc1 and Sec7-mKate in their mutant backgrounds. Regarding overlap of Sec7-RFP and GFP-Tlg1, Saimani et al. (2017) report 50% colocalization (PMID: 28521960)." (Reviewer 3).

We would like to thank you and the reviewers for taking the time and effort to evaluate our manuscript. We have resubmitted the manuscript with a change to the title and Abstract that reflects the primary conclusions of the manuscript that all reviewers agree with. We also did our best to completely scrub “the early endosome” description from the conclusions of our experimental work and therefore we have substantially revised the manuscript again. We simply state that the perturbations cause a defect in recycling and revised the Introduction to provide the following definition for the recycling pathway. “However, given the uncertainty in the nature of these compartments marked by Tlg1 and Sec7 (early endosome, TGN, or a hybrid of these organelles), we use the term “recycling” to indicate movement of GFP-Snc1 from the endocytic pathway to the exocytic pathway”. For the localization and action of COPI, we conclude it is being recruited to compartments of the recycling pathway rather than saying it is budding vesicles from the early endosome. The Discussion was shortened by eliminating sections discussing the potential role of COPI on early endosomes.There are a few places in the Introduction and Discussion where we describe published data that used this early endosome descriptor to place our work in context of published work (Introduction) and to provide what we firmly believe is the most logical conclusion of the study (Discussion). We have performed a number of additional experiments to further address the nature of these compartments and trafficking pathways as suggested by the reviewers, but hope to publish these results in a separate manuscript as they have moved well beyond the scope of the current study and do not alter our conclusions.